# Roles of histone chaperone Nap1 and histone acetylation in regulating phase-separation of nucleosome arrays

Jia Gao [1], Hongyun Li[1], Song Tan [2], Ruobo Zhou [1,2] & Tae-Hee Lee [1] ✉

Chromatin condensation is dynamically regulated throughout the cell cycle and plays key roles in modulating gene accessibility. The DNA-histone dynamics in the nucleosome are central to the regulation mechanisms of chromatin condensation, which remain poorly understood. Employing fluorescence recovery after photobleaching, optical super-resolution imaging, and microrheology with optical tweezers, we investigated the roles of various parameters in regulating phase-separation of 12-mer nucleosome arrays. Here, we show that histone H4 tail lysine residues are the main drivers of liquid-liquid phase separation of nucleosome arrays. We also show that the condensed liquid-like droplets comprise a mobile fraction and a relatively immobile structural scaffold. Histone chaperone Nap1 and histone H3 tail acetylation enhance DNA-histone dynamics within this scaffold, thereby lowering the overall viscosity of the droplets. These results suggest that histone chaperone and histone H3/H4 tails play critical roles in regulating chromatin condensation and gene accessibility in condensed chromatin.

The eukaryotic cell nucleus consists of two main compartments, one of which is the chromatin that contains the genome. Although not separated by a physical barrier, chromatin exists in its own phase, while the nucleoplasm can also contain other membrane-less compartments and liquid-like droplets[1–3]. Chromatin is composed of arrays of genomic DNA, histone proteins, and various other chromatin-associated proteins. The fundamental DNA packaging unit in chromatin is the nucleosome, which is ~150 bp DNA wrapped around an octameric histone core containing two H2A-H2B dimers and one $(H3\text{-}H4)_2$ tetramer[4]. The concentration of nucleosomes in a cell nucleus is in the range of a few tens to hundreds of μM[5]. The interactions between DNA and histones, both within and across nucleosomes, are central to the often complex mechanisms of gene regulation at multiple levels[6–8]. One way to regulate such dynamic genome activities is through histone post-translational modifications[9–11]. Another involves enzymes that act on chromatin, such as histone chaperones and chromatin remodelers[12–15]. Efforts at multiple levels both in vivo and in vitro are being expended on elucidating the mechanisms of how these factors help regulate genome

activity[16–22]. In particular, in vitro approaches with highly refined systems have made valuable contributions to deciphering complex changes at the nucleosome and DNA-histone levels without interference from unknown factors[23–26]. Many such approaches employ observing and controlling nucleosomes at the single-nucleosome level, either in individual nucleosomes or in nucleosomes linked in an array. However, these systems lack a fundamental aspect of chromatin where nucleosomes are densely packed at a high concentration. Once the concentration of nucleosomes or nucleosome arrays is elevated to more than a few tens to hundreds of nM under a physiological ionic condition, they undergo liquid-liquid phase separation (LLPS) to form liquid-like droplet condensates[27–29]. The concentrations of nucleosomes within these droplets match those in vivo[27]. While the mechanism of condensed droplet formation remains under active investigation, it should involve both short- and long-range inter-nucleosomal interactions, which are likely relevant to chromatin condensation in vivo. A previous paper reported that the droplets are more gel-like than liquid-like, although the shape of many droplets is still quite spherical, suggesting that the droplets

[1]Department of Chemistry, The Pennsylvania State University, University Park, PA, USA. [2]Department of Biochemistry and Molecular Biology, The Pennsylvania State University, University Park, PA, USA. ✉e-mail: txl18@psu.edu

begin in a liquid-like state and age into a gel-like state[30]. Moreover, the gel-like nature of the droplets has been largely disproved by a recent report[27]. Nevertheless, these condensates, especially those made of nucleosome arrays, are excellent platforms for investigating the physical and biological properties and functions of nucleosomes and chromatin-acting enzymes in a chromatin-like environment.

Histone N-terminal tail acetylations on H3 or H4 have been implicated in weakened intra- and inter-nucleosomal DNA-histone interactions[31–38], thereby modulating the strength of chromatin condensation at both the nucleosome and chromatin levels. Histone H3 tail acetylation is more strongly associated with intra-nucleosomal DNA-histone compaction, while H4 tail acetylation is more strongly coupled to inter-nucleosomal chromatin compaction[31–36]. Previous reports suggested that histone acetylation generally inhibits phase separation of nucleosomes or nucleosome arrays, or that it reduces droplet size or causes them to dissolve[27,39]. However, the effects of specific histone tail acetylations on the condensation behavior of nucleosome arrays remain poorly understood. Another important mediator of DNA-histone interactions is the class of histone chaperones. Histone chaperones contain acidic residues that can compete for histone binding against DNA[40]. Histone chaperone Nap1 can bind both H2A-H2B and (H3-H4)$_2$ in a stoichiometric manner[41,42]. Nap1 mediates DNA-histone interactions to bring a random mixture of DNA and histones to their thermodynamically favored state which is the nucleosome[43]. Despite its important role in regulating nucleosome stability, it has never been reported how Nap1 functions in the condensed phase of nucleosomes. In particular, it remains unknown on what timescale Nap1 enables or facilitates DNA-histone dynamics to eventually stabilize dynamic nucleosomes in a condensed phase. To this end, we investigated how histone H3 or H4 acetylation and histone chaperone Nap1 regulate the formation and the properties of liquid-like chromatin condensates spontaneously formed by LLPS of 12-mer nucleosome arrays.

Here, we show that H4 tail acetylation inhibits chromatin condensation while H3 tail acetylation mimic (lysine residues mutated to glutamine) makes the condensed chromatin droplets more fluidic, based on bright-field microscopic imaging, fluorescence recovery after photobleaching (FRAP) observations, and microrheological measurements with optical traps. We also show that histone chaperone Nap1 lowers the viscosity of the chromatin droplets, while it significantly elevates the nucleosome concentration within a condensate. Optical super-resolution images reveal that condensed droplets contain both mobile and relatively immobile structural scaffold nucleosome arrays. The microrheology results indicate that the effects of the H3 tail acetylation mimic and Nap1 are exerted specifically on the nucleosome arrays associated with the structural scaffold. In summary, our study reports that histone H4 tail acetylation inhibits chromatin condensation, while histone H3 tail acetylation and histone chaperone Nap1 render a more fluidic yet stable structure of condensed chromatin, suggesting their roles in regulating gene accessibility at a chromatin level.

## Results

### Nucleosome arrays form liquid-like droplet condensates

Nucleosome arrays were reconstituted with human histone octamers and a DNA template containing 12 repeats of the Widom 601 nucleosome positioning sequence with a 25 bp linker length (Supplementary Fig. 1). Proper saturation of reconstituted nucleosome arrays was confirmed by native gel electrophoresis, transmission electron microscopy (TEM), analytical ultracentrifugation, a restriction enzyme digestion assay, and an SDS-PAGE assay (Supplementary Figs. 2–4). These nucleosome arrays at 200 nM undergo liquid-liquid phase separation (LLPS) into droplet condensates when the ionic environment is set at 150 mM NaCl (Fig. 1a). This result further confirms the proper assembly of nucleosome arrays. A previous report has shown that the addition of various physiological monovalent salts (e.g.,

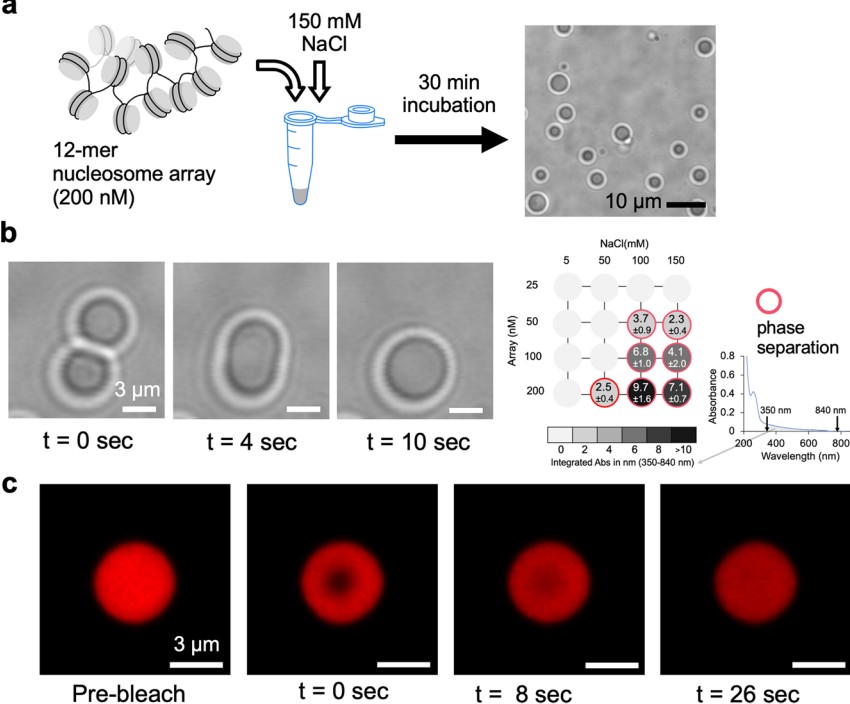

**Fig. 1 | Formation of liquid-like condensates by nucleosome arrays at a physiological salt concentration of 150 mM NaCl. a** Experimental scheme to form chromatin condensates: Dodecamer nucleosome arrays at 200 nM incubated in a buffer containing 150 mM NaCl result in condensate droplets. The scale bar is 10 μm. **b** Droplets fuse with each other in a few seconds timescale, confirming their liquid-like nature. The experiment was repeated more than 10 times with three or more samples prepared on three or more different days. The scale bars are 3 μm. **c** Fluorescence recovery after photobleaching (FRAP) also confirms the liquid-like nature of the droplets. The scale bars are 3 μm. The experiment was repeated more than 10 times with three or more samples prepared on three or more different days.

150 mM NaCl) to 12-mer nucleosome arrays at a similar level of concentration results in liquid-like droplet condensates[27]. Fusion of droplets on a few-second timescale confirms the liquid-like nature of these condensates (Fig. 1b). To survey the extents of droplet formation under a range of conditions, the UV-Vis absorbances of the arrays were measured at varying array concentrations (25, 50, 100, and 200 nM) and NaCl concentrations (5, 50, 100, and 150 mM). Droplets scatter light at wavelengths according to their sizes. The scattering extent was measured as a UV–Vis absorbance spectrum within the range of 350–840 nm. Non-zero detectable absorbances in this range confirm the formation of droplets down to a few hundred nanometers in size. A time series of changes in the UV-Vis spectrum during the first 30 min of droplet formation is shown in Supplementary Fig. 5. As the changes are not monotonic increase or decrease at a specific wavelength in this range, we used the integrated UV-Vis absorbance (350–840 nm) after 30 min of droplet formation as a measure of the abundance of the droplets of various sizes. The measured droplet abundances are presented as a phase diagram (Fig. 1b). Using fluorescently labeled histone H4 at E63C with Alexa Fluor 647, we found that the concentrations of the nucleosomes in the chromatin droplets are $326 \pm 26$ μM, in excellent agreement with a previous report[27]. The fast recovery of fluorescence after photobleaching (Fig. 1c) also confirms the efficient and constant motions of nucleosomes inside the condensed droplets. The condensates remain liquid-like at least for 2 h after adding 150 mM NaCl.

## Histone H4 tail lysine residues are the main drivers of LLPS

Histone acetylations have been coupled to the weakening of phase separation of nucleosomes and nucleosome arrays[27,39]. In order to investigate the roles of H3 and H4 tails and their acetylation states in regulating LLPS of nucleosome arrays, we prepared four types of nucleosome array samples containing a modified histone core with tailless H3 (gH3), tailless H4 (gH4), an H3 tail acetylation mimic (H3KQ), and an H4 tail acetylation mimic (H4KQ). In H3KQ and H4KQ, the lysine residues in the N-terminal tail (K4, K9, K14, K18, K23, and K27 of H3 and K5, K8, K12, K16, and K20 of H4) are mutated to glutamine to mimic acetylation[44–48]. Among these four samples, only the H3KQ arrays result in chromatin droplets (Fig. 2a and Supplementary Fig. 6a). The H4KQ arrays show no sign of droplet formation up to 500 nM array concentration and 200 mM NaCl concentration for 2 h, while the gH3 and gH4 arrays result in irregularly shaped aggregates (Fig. 2a). Note that gH4 arrays form only a very small amount of aggregates that are not easy to find on a microscope. The phase diagrams (Figs. 1b and 2b) clearly show less efficient droplet formation with H3KQ arrays or H3 acetylated arrays than with WT arrays at given array and NaCl concentrations, and no droplet formation with H4KQ arrays. The phase diagram of gH3 arrays was constructed in the same way as the other phase diagrams (i.e., integrated scattering strengths in the range of 350 – 840 nm). The small amount of aggregates formed with gH4 arrays is not detectable with a UV-Vis spectrometer. We also confirm the formation of aggregates that are distinct from droplets with UV-Vis measurements of the supernatant and the bottom fraction of the aggregate and droplet samples upon centrifugation (Supplementary Fig. 7). These results suggest that H3 and H4 tails play distinct roles in chromatin phase separation and that the H4 tail lysine residues are the main drivers of chromatin LLPS.

To further confirm the effect of H4KQ, we tested if H4 tail acetylation with histone acetyltransferase (HAT) also inhibits LLPS. We employed the Piccolo NuA4 complex that acetylates histone H4 tail lysine residues[49,50]. We acetylated unmodified (WT) arrays with Piccolo NuA4 and tested phase separation to observe no droplet formation under various conditions (see the phase diagram in Fig. 2c). Furthermore, upon adding Piccolo NuA4 to chromatin droplets already formed with WT nucleosome arrays, we observed gradual dissolution of the droplets (Fig. 2c). The fast enzyme access to histone tails further

validates that droplets are liquid-like and can freely exchange materials with the outer environment. Incubation with Piccolo NuA4 and coenzyme-A (Co-A) instead of acetyl Co-A (Ac-CoA) induces no difference in the droplets, confirming that the change is acetylation-dependent (Fig. 2c). Upon a closer look at the droplets, we observed crumbling from the inside, loss of the round outer fringe, and shrinkage of the droplet size within 10–15 min (Supplementary Fig. 8). We also tested H3 tail acetylation in situ with the Ada2/Ada3/Gcn5 complex[51,52]. As expected, no changes were observed in droplet formation under various conditions, further validating the observation with the H3KQ acetylation mimic (Fig. 2d). These results confirm that the H4 tail lysine residues are the main drivers of chromatin LLPS regardless of the nucleosome concentration up to a physiological level of a few hundred μM.

## Histone chaperone Nap1 dissolves chromatin aggregates

The gH3 and the gH4 array samples result in solid or gel-like condensates that do not fuse (Fig. 2a). In particular, gH3 arrays induce large aggregates, while gH4 arrays do so to a much lesser extent. We hypothesize that the aggregation is caused by partial nucleosome disassembly upon nucleosome destabilization due to the lack of the H3 tails. Tailless H3 induces nucleosome destabilization although it does not result in spontaneous disassembly of mono-nucleosomes[36,53]. However, nucleosomes destabilized at a mono-nucleosomal level may lead to their disassembly at least partially when they are condensed. This is because the transiently exposed histone core by the destabilized DNA wrapping would be prone to random histone-DNA interactions in a condensed phase, leading to nucleosome disassembly.

To test this hypothesis, we added histone chaperone Nap1 to the aggregates formed with gH3 arrays. We observed that upon the addition of Nap1 at a molar ratio of 1:2 (nucleosome:Nap1), the solid or gel-like aggregates dissolve away within a few minutes (Fig. 3a). When Nap1 is added to gH3 arrays at a molar ratio of 1:2 (nucleosome:Nap1) before inducing LLPS, the aggregate formation was significantly reduced (Fig. 3b and Supplementary Fig. 9). No change in LLPS was observed when Nap1 was added to the condensates formed with WT arrays at a molar ratio up to 1:8 (nucleosome:Nap1) (Fig. 3c). These results support our hypothesis of random histone-DNA interactions and potential nucleosome disassembly in condensed chromatin containing tailless H3. The fast action of Nap1 with the pre-formed aggregates strongly supports that these aggregates are more permeable gel-like than solid.

To examine the extent of nucleosome unwrapping in gH3 arrays, we employed a restriction enzyme digestion assay with two restriction enzymes HinfI and BsrBI (Supplementary Fig. 10). The 601 nucleosome sequence contains a HinfI restriction site at the $5^{th}$–$10^{th}$ nt from the entry and a BsrBI restriction site at the $23^{rd}$ to $28^{th}$ nt from the entry. Partially unwrapped nucleosomes at the nucleosome termini will make the HinfI site accessible and digested by the enzyme while more completely disassembled nucleosomes will make the BsrBI site accessible and digested. Our digestion assay shows that gH3 arrays are partially digested by HinfI while they are not digested by BsrBI, supporting partial unwrapping of nucleosomes in gH3 arrays. Aggregated gH3 arrays are digested even more, supporting potential partial disassembly of the nucleosomes in the aggregates. Upon Nap1 treatment, which dissolves most aggregates, we observe reduced HinfI digestion, indicating that some of the partially unwrapped nucleosomes are re-wrapped in the presence of Nap1.

## H3KQ and Nap1 make chromatin condensates more fluidic

The histone H3 tail lysine residues play important roles in stabilizing nucleosomes by interacting with DNA[31,32,54,55]. Therefore, their acetylation should affect the dynamics of DNA-histone interactions at both the nucleosome and the chromatin levels. To test this hypothesis, we measured the dynamics of fluorescence recovery after photobleaching (FRAP) within the droplets formed with WT and H3KQ arrays. We

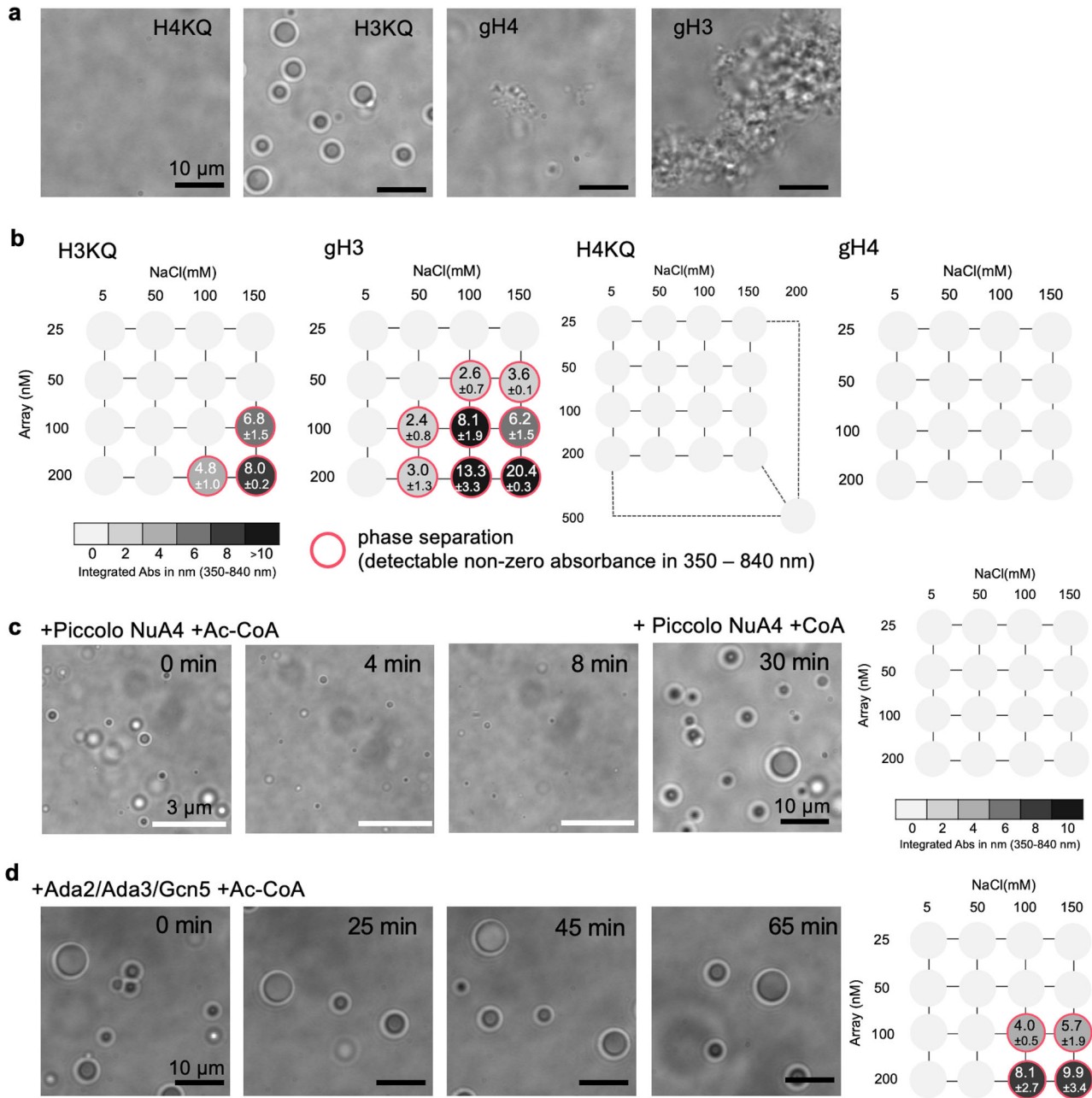

**Fig. 2 | Roles of Histone H4 and H3 tails in regulating phase separation of nucleosome arrays. a** Nucleosome arrays with the H4 tail acetylation mimic (H4KQ) do not form condensates, but those with the H3 tail acetylation mimic (H3KQ) form spherical phase-separated droplets. Nucleosome arrays with H4 or H3 tails truncated (gH4 or gH3) form solid or gel-like aggregates. The scale bars are 10 μm. The pictures represent 6 or more repeated observations made with two or more samples prepared on two or more different days. More images are available in Supplementary Fig. 11. **b** Phase diagrams of WT (unmodified), H3KQ, gH3, H4KQ and gH4 arrays show less efficient droplet formation with H3KQ than WT arrays and no droplet formation with H4KQ under the conditions tested. The small amount of aggregates formed with gH4 arrays is not detectable with a UV-vis spectrometer. Two or more measurements with two or more different samples were made. **c** In situ H4 tail acetylation by Piccolo NuA4 histone acetyltransferase dissolves the phase-separated droplets already formed with nucleosome arrays with no modifications, confirming the effect shown in **a**. Incubation with the enzyme and

coenzyme A (CoA) instead of acetyl-CoA (Ac-CoA) does not dissolve the condensates, confirming that the effect is acetylation-dependent. The white scale bars are 3 μm. The pictures represent 4 or more repeated observations made with two or more samples prepared on two or more different days. The phase diagram represents no phase separation behavior of WT arrays acetylated by Piccolo NuA4. For phase diagram, two or more measurements with two or more different samples were made. **d** In situ H3 tail acetylation with Ada2/Ada3/Gcn5 histone acetyltransferase does not affect phase separation of nucleosome arrays. The scale bars are 10 μm. The pictures represent 4 or more repeated observations made with two or more samples prepared on two or more different days. The phase diagram represents the phase separation behavior of WT arrays acetylated by Ada2/Ada3/Gcn5, clearly showing a similar phase separation behavior to H3KQ arrays. For phase diagram, two or more measurements with two or more different samples were made. More images are available in Supplementary Fig. 6.

employed fluorescently labeled histone H4 at E63C (Tet) and H2B at T115C (Di)[56]. We observed at least 2-fold faster recovery on average with H3KQ arrays than with WT arrays in both Tet- and Di-labeled cases (Fig. 4a and Supplementary Fig. 11, and Supplementary Table 1). The

FRAP recovery fractions are comparable in the two cases with no significant difference. These results indicate that nucleosome motions are facilitated in H3KQ droplets, suggesting that DNA-histone dynamics are enhanced at the length and time scales of nucleosome

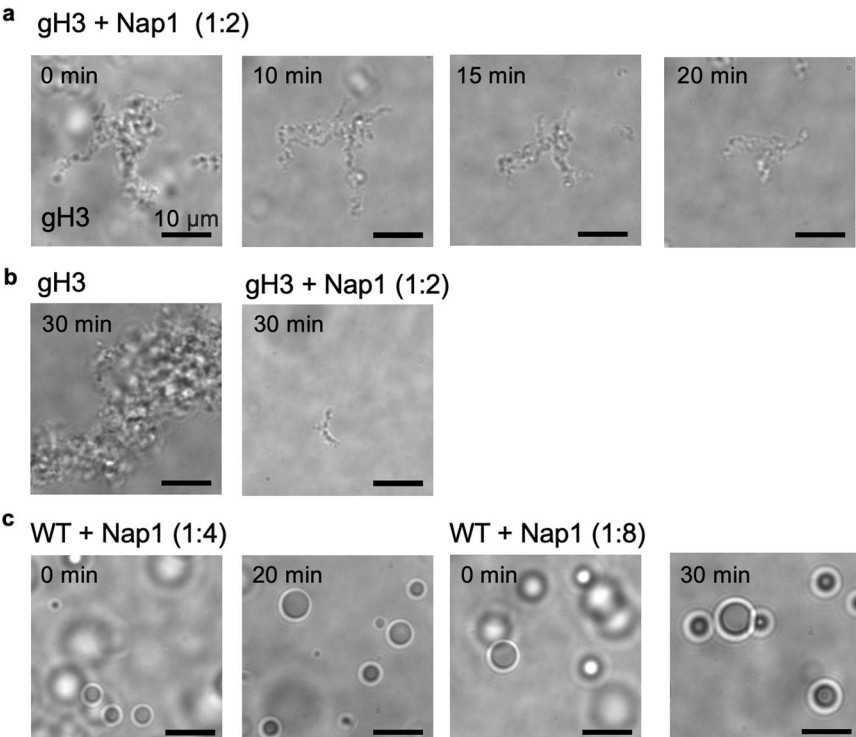

**a** gH3 + Nap1 (1:2)

0 min gH3 10 μm

10 min

15 min

20 min

**b** gH3 gH3 + Nap1 (1:2)

30 min

30 min

**c** WT + Nap1 (1:4) WT + Nap1 (1:8)

0 min

20 min

0 min

30 min

**Fig. 3 | Dissolution of solid or gel-like chromatin aggregates by histone chaperone Nap1. a** Histone chaperone Nap1 dissolves pre-formed chromatin aggregates with nucleosome arrays containing tailless histone H3 (gH3) at a molar ratio of 1:2 (nucleosome:Nap1). The pictures represent two repeated observations made with two samples prepared on two different days. **b** Histone chaperone Nap1 pre-mixed with gH3 nucleosome arrays at a molar ratio of 1:2 (nucleosome:Nap1) before adding 150 mM NaCl significantly inhibits the formation of solid or gel-like chromatin aggregates. The pictures represent two repeated observations made with two samples prepared on two different days. More images are available in Supplementary Fig. 9. **c** The effect of Nap1 presented in (**a**) is not observed with spherical droplets formed with arrays with no modification (WT) up to a molar ratio of 1:8 (nucleosome:Nap1) for 30 min. All scale bars are 10 μm. The pictures represent two repeated observations made with two samples prepared on two different days.

motions. This effect is likely due to weakened interactions between DNA and H3 tail lysine residues. To further confirm the effect, we employed histone acetyltransferase (HAT) Ada2/Ada3/Gcn5 to acetylate H3. We first verified acetylation of arrays in condensates when incubated with this HAT complex (Supplementary Fig. 12). We then repeated the FRAP assay with the droplets containing arrays that are H3-acetylated (H3ac) by this HAT complex (Fig. 4a). The droplets display considerably faster FRAP recovery than those containing WT arrays, while the recovery fraction remains unchanged. The HAT complex has a similar acetylation activity to the entire SAGA complex, which acetylates mainly H3 tail lysine residues, including K9, K14, K18, and K23[57–59]. However, the exact locations and extents of acetylation by Ada2/Ada3/Gcn5 are yet to be clearly defined. Consequently, we cannot attribute the observed effect of H3 acetylation to specific lysine residues or to the degree of their modification. Nonetheless, these results confirm that H3 acetylation considerably enhances DNA-histone dynamics in condensates. Combined with the results from gH3 arrays that aggregate once condensed, these observations suggest that the H3 tail, even when the lysine residues are acetylated and lose their positive charges, still plays a significant role in stabilizing liquid-like chromatin condensates.

To test the effect of Nap1 on regulating DNA-histone dynamics in condensed chromatin, we monitored the FRAP recovery dynamics within the droplets formed with WT arrays in the presence and absence of Nap1 (Fig. 4a). According to the observations, Nap1 makes the chromatin droplets more fluidic in the relevant length and time scales in both Tet- and Di-labeled cases. Spontaneous DNA-histone dynamics on a few milliseconds timescale result in reversible and repetitive partial disassembly and reassembly of nucleosomes. Our results suggest that Nap1 facilitates and stabilizes these microscopic dynamics to

induce a macroscopic change in the condensed nucleosome motions on a few to a few tens of seconds timescale[60]. To further support this mechanism, we monitored the FRAP recovery dynamics of Di-labeled array condensates at varying ratios of nucleosome:Nap1 at 1:0.1, 1:2, 1:4, and 1:8 (Supplementary Fig. 13). The results show that a stoichiometric amount of Nap1 is required to induce a significant change in the FRAP-recovery time in condensates. A catalytic amount of Nap1 at 1:0.1 does not result in any noticeable changes in the dynamics, suggesting that the enhanced FRAP recovery is the result of Nap1 facilitating transient and repetitive partial disassembly/reassembly of the nucleosome in the H2A-H2B region. The observation that the enhancement is plateaued at 1:4 nucleosome:Nap1 ratio also supports this mechanism as Nap1 interacts mainly with H2A-H2B unless the nucleosome is considerably disassembled[43]. We also support this conclusion by showing that Nap1 is enriched in the condensed phase according to the results from an SDS-PAGE assay (Supplementary Fig. 14). These results suggest that Nap1's role in making nucleosomes more dynamic in condensed chromatin is localized where Nap1 directly interacts with nucleosomes.

To examine the mechanism of FRAP recovery, we compared the dynamics from the two different labeling positions which are H4E63C (Tet) and H2BT115C (Di). The Tet-labeled arrays show a slower recovery time than the Di-labeled ones in the WT, the Nap1, and the H3ac cases (Fig. 4a). The difference is not significant in H3KQ arrays, likely because the recovery times are already too short to reliably gauge any differences at the resolution of our measurements. We propose that this difference is due to spontaneous histone H2A-H2B dimer exchange between nucleosomes[61]. In addition, the FRAP recovery time in the Nap1 Di case is nearly two-fold faster than that in the WT Tet/Di case, supporting a role for Nap1 in facilitating H2A-H2B diffusion within

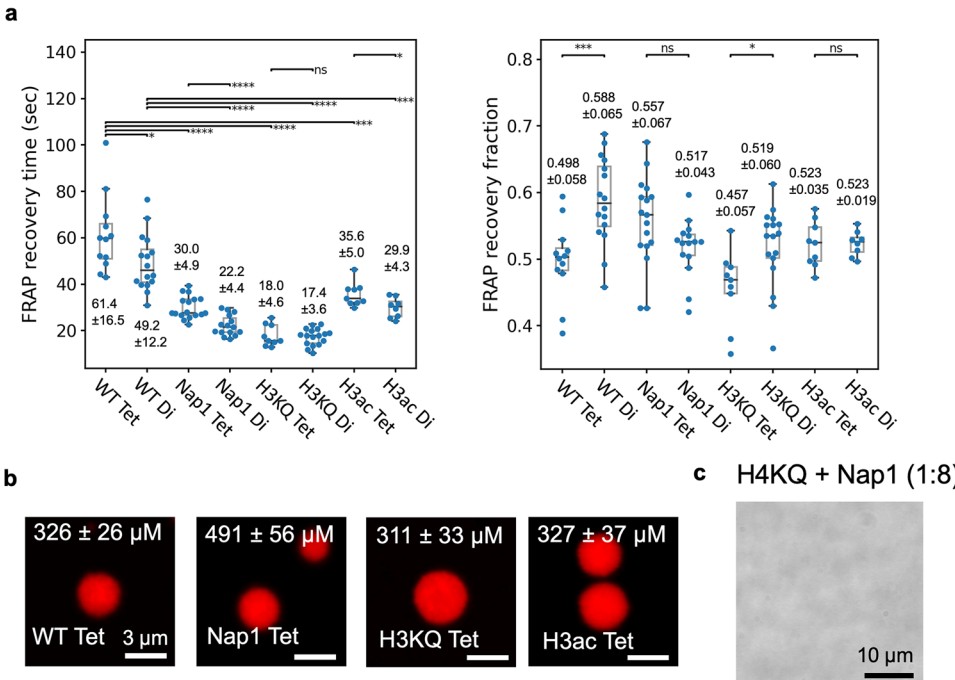

**Fig. 4 | Enhanced nucleosome dynamics in condensed liquid-like droplets by histone chaperone Nap1, H3 tail acetylation mimic, and H3 acetylation by Ada2/Ada3/Gcn5. a** FRAP recovery times and recovery fractions are plotted for the condensates formed with unmodified nucleosome arrays (WT) labeled at histone H4 E63C (Tet) and at H2B T115C (Di), WT arrays in the presence of Nap1 (Nap1), H3KQ arrays (H3KQ), and H3 acetylated arrays by Ada2/Ada3/Gcn5 (H3ac). The significances are from two-sided Student's $t$ test (*: $p \leq 0.05$, **: $p \leq 0.01$, ***: $p \leq 0.001$, ****: $p \leq 0.0001$, ns: not significant). The p-values for the recovery times to compare WT Di, Nap1 Tet, H3KQ Tet, and H3ac Tet with WT Tet are 0.0435, 2.96 $\times 10^{-5}$, 8.16 $\times 10^{-7}$, and 1.76 $\times 10^{-5}$, respectively, those to compare Nap1 Di, H3KQ Di, and H3ac Di with WT Di are 1.00 $\times 10^{-7}$, 1.20 $\times 10^{-8}$, and 1.38 $\times 10^{-5}$, respectively, and those to compare Nap1 Tet with Nap1 Di, H3KQ Tet with H3KQ Di, and H3ac Tet with H3ac Di are 5.12 $\times 10^{-5}$, 0.747, and 0.0225, respectively. The p-values for the recovery fractions to compare WT Tet with WT Di, Nap1 Tet with Nap1 Di, H3KQ Tet

with H3KQ Di, and H3ac Tet with H3ac Di are 0.000722, 0.0568, 0.0202, and 0.950, respectively, respectively. The sample sizes $n$ = 12, 16, 17, 15, 9, 17, 9, and 8, respectively for the WT Tet, WT Di, Nap1 Tet, Nap1 Di, H3KQ Tet, H3KQ Di, H3ac Tet, and H3ac Di cases. The sample size refers to the number of distinct droplets tested in each case. The samples and the measurements were made on at least two different days. The values marked on the box plots represent mean ± standard deviation. Box plot elements are center line (median); box limits (upper and lower quartiles); whiskers (1.5x interquartile range). Source data are provided as a Source Data file. **b** Concentrations of nucleosome arrays in droplets are shown. The sample sizes $n$ = 15, 42, 21, and 8, respectively, for the WT Tet, Nap1 Tet, H3KQ Tet, and H3ac Tet cases. The samples and the measurements were made on at least two different days. **c** Nap1 (1:8 nucleosome:Nap1 molar ratio) does not result in droplet formation with H4KQ arrays up to 2 h.

condensed chromatin possibly by catalyzing spontaneous H2A-H2B exchange between nucleosomes.

## Facilitated dynamics is not due to decreased concentration

We further validated the roles of H3 tail lysine residues and Nap1 in facilitating chromatin dynamics in a condensed phase by showing that the nucleosome concentration within a droplet does not decrease in these cases. We measured the concentrations of the nucleosomes in the four cases (WT, Nap1, H3KQ, and H3ac) with the Tet-labeled samples according to the fluorescence intensities from the confocal images (Fig. 4b) and the calibration with the fluorophore (Supplementary Fig. 15). We found that the concentrations stayed constant within error between the WT ($326 \pm 26 \mu M$), the H3KQ ($311 \pm 33 \mu M$), and H3ac ($327 \pm 37 \mu M$) cases, confirming that the H3 acetylation effect on making chromatin more dynamic is not due to decreased nucleosome density in the droplets. We also found that the nucleosome concentration increases significantly upon the addition of Nap1 from $326 \pm 26$ (WT) to $491 \pm 56 \mu M$ (Nap1). The Di-labeled arrays also show a concentration increase ($371 \pm 36$ vs $413 \pm 24 \mu M$, see Supplementary Fig. 16). Although the level of increase is smaller, the difference is significant according to the student's t-test (two-sided $p \leq 0.0001$, see Supplementary Fig. 11). The increased concentration of Di-labeled nucleosomes is the same as the increased concentration of the Tet-labeled nucleosomes within error at the $1\sigma$ confidence interval ($413 \pm 24$ vs $491 \pm 56 \mu M$). The increased concentration of nucleosomes

by Nap1 is likely due to the elevated thermodynamic stability of nucleosomes in the presence of Nap1. This function of Nap1 in elevating nucleosome concentration in condensed chromatin has never been reported. To this end, we tested if the droplet formation is induced specifically by the inter-nucleosomal interactions between and within nucleosome arrays that are mediated by histone H4 tail lysine residues, or generally by an increased nucleosome concentration that can be caused by either H4-mediated inter-nucleosomal interactions or various other factors such as Nap1. When Nap1 was added to H4KQ arrays at up to a 1:8 (nucleosome:Nap1) molar ratio, no droplet formation was observed up to 2 h. This result further confirms that H4 tail lysine residues are the main drivers of chromatin LLPS, regardless of the nucleosome concentration or stability within a condensate.

## High resolution imaging reveals a static structural scaffold

The FRAP results (Fig. 4a) indicate that the fluorescence recovery is limited to ~50 to ~60 %. This observation led us to a hypothesis that condensed chromatin droplets contain a network of nucleosome arrays that form a relatively immobile structural scaffold of the droplet, moving too slowly to be captured during our FRAP measurements (up to 216 s). To test this hypothesis, we used stochastic optical reconstruction microscopy (STORM) to image nucleosomes that are relatively static on the time scale of 10 s in droplets (Fig. 5 and Supplementary Fig. 17). This time scale is similar to the droplet fusion time

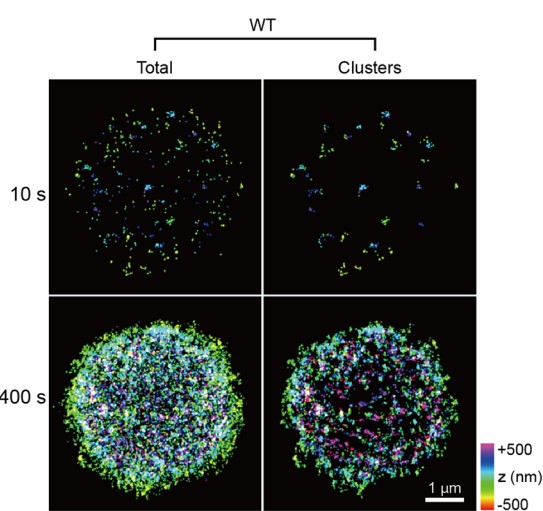
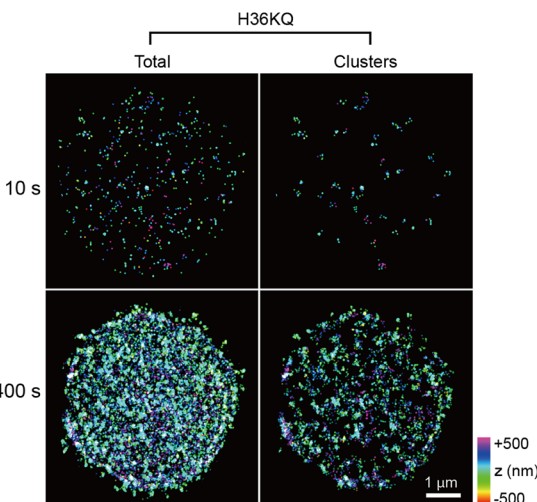
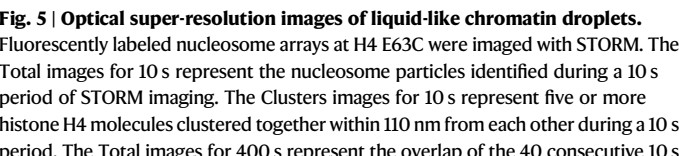

**Fig. 5 | Optical super-resolution images of liquid-like chromatin droplets.**
Fluorescently labeled nucleosome arrays at H4 E63C were imaged with STORM. The Total images for 10 s represent the nucleosome particles identified during a 10 s period of STORM imaging. The Clusters images for 10 s represent five or more histone H4 molecules clustered together within 110 nm from each other during a 10 s period. The Total images for 400 s represent the overlap of the 40 consecutive 10 s

Total images. The Clusters images for 400 s represent the overlap of the 40 consecutive 10 s Clusters images. These STORM images of both WT and H3KQ arrays confirm a structural scaffold of nucleosome arrays that are relatively immobile on the timescale of 10 s. The z-position of the localized nucleosomes are color coded as shown in the color scale. The analysis was performed for more than five droplets per case. Five more images in each case are available in Supplementary Fig. 17.

scale (typically <10 s for droplets of a few μm diameter) so that we can observe both relatively mobile and immobile fractions. We took fluorescence images of chromatin droplets containing Tet-labeled nucleosome arrays and constructed their STORM images (see Methods). The 10 s Total STORM images in Fig. 5 show all fluorescent nucleosome arrays detected during a 10 s imaging period. Within each STORM image, we identified Clusters of nucleosome arrays that remained immobile during this period within a 110 nm radius, approximately the contour length of a 12-mer nucleosome array. The 10 s Clusters STORM images in Fig. 5 display these immobile arrays. Comparing the STORM images of Total and Clusters in Fig. 5 reveals that many H4 molecules remain relatively static during a 10 s period, while the others are more mobile with a motion speed at or above our 100 Hz image acquisition rate. Forty 10 s STORM images were overlaid to generate Total and Clusters views, representing all nucleosomes and relatively static nucleosome clusters, respectively, identified over the full 400 s imaging session (Fig. 5). From these STORM images, it is evident that the entirety of the droplet interior is full of relatively static nucleosomes on a 10 s timescale represented by the dots on a Clusters image. Such an immobile fraction is evident in mixing the contents of two droplets during a fusion event (Supplementary Fig. 18). During fusion, the boundaries of two droplets first fuse to form one elongated droplet with two distinct interior phases, and then their contents mix in the next few seconds. It is also evident that there are mobile nucleosomes with a motion speed at or above our 100 Hz image acquisition rate on a 10 s timescale shown as the dots on a Total image which do not appear on the corresponding Clusters image in both the WT and the H3KQ cases. Overall, STORM imaging confirms that condensed chromatin droplets contain both mobile and relatively immobile scaffolding nucleosomes on a 10 s timescale.

**Microrheology confirms low viscosity with Nap1 or H3KQ**
FRAP recovery times provide insights into how dynamically nucleosomes move within a droplet. FRAP recovery within a droplet reflects the motions of nucleosomes from the mobile fraction of the condensate, which pass through both mobile and immobile nucleosomes and arrays. The motions encounter friction due to collisions, most strongly with other nucleosome arrays. Such friction results in slower diffusion, which can be interpreted as a higher viscosity of the

diffusion medium. The FRAP recovery times are a convolution of multiple parameters affecting diffusion, such as the viscosity of the solvent and various interactions of the diffusing body with other components in the diffusion medium (e.g., collisions, transient association, and dissociation). The Stokes-Einstein relation between diffusion and viscosity does not accurately reflect such complex dynamics. Moreover, polymeric materials often display viscoelastic behavior that depends on the rate of motion. Therefore, a simple convolution of these factors into FRAP recovery times would not properly capture the nucleosome dynamics and their changes due to modified histones and the presence of Nap1. To this end, we measured the viscoelasticity of chromatin droplets using an optical tweezers setup. We brought a polystyrene bead (1 μm diameter) into a droplet (>4 μm diameter) and sinusoidally oscillated the bead with a ± 0.2 μm amplitude (Fig. 6a). We employed a Burgers model, a simple linear viscoelasticity model, with two Maxwell components to interpret the data[62]. The drag and the attenuation in the bead motion at various oscillation frequencies contain all the relevant pieces of information (Fig. 6a). The storage and loss moduli at each frequency can be extracted from these bead motions. Fitting the frequency dependence of the moduli results in the relaxation times and the viscosities (Fig. 6b). A single Maxwell component did not fit the data. The two-component fitting worked well, and the results are shown in Fig. 6c. The faster relaxation component has a relaxation time of a few ms which is 2–3 orders of magnitude shorter than the slower component, which shows a relaxation time of a few hundred ms to a few seconds. The viscosity of the faster component is 2–3 orders of magnitude lower than that of the slower component and only 2–3 times higher than that of water (0.001 Pa·s at 25 °C). Therefore, the faster component is likely due to the dynamics of the mobile arrays freely diffusing within a droplet. The faster component should result in diffusion resistance from the mobile arrays, while the slower component exerts resistance from the structural scaffold. The faster relaxation would be governed mainly by intra-array conformational dynamics of mobile arrays during diffusion while the slower relaxation would be governed mainly by rearrangements of the intra- and inter-array networks in the scaffold. The 2–3 orders of magnitude difference in their timescales is well aligned with the fact that the network reconfiguration of the scaffold arrays should be much slower than the conformational changes of mobile arrays[63]. Under the

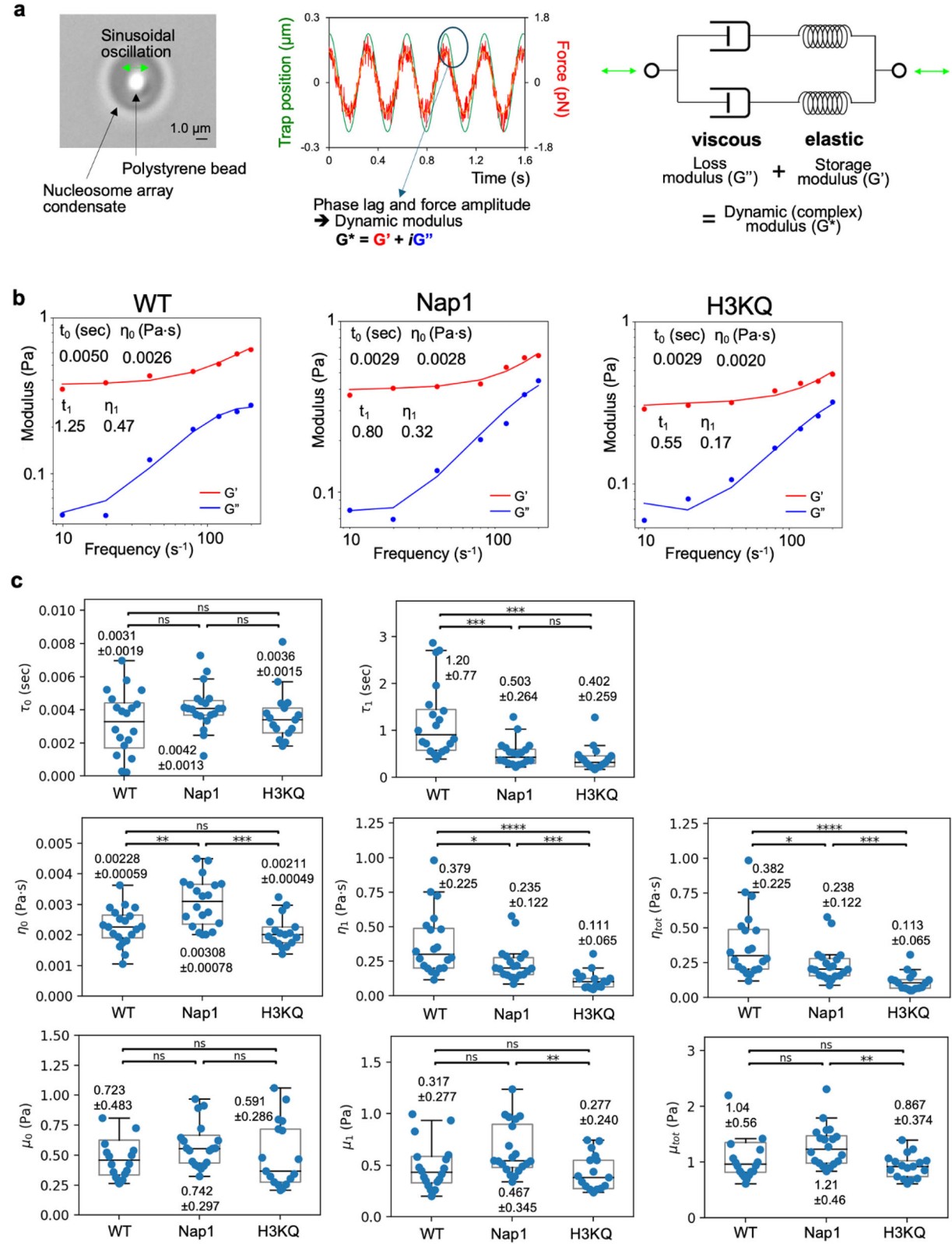

condition of 2-3 orders of magnitude difference in the viscosity, the network reconfiguration time is approximately the relaxation time of the slower component which is a few hundred ms to a few seconds. Processes that are faster than this timescale will experience both viscous and elastic properties of the chromatin while slower processes will experience mostly the viscous nature with the viscosity values shown in Fig. 6c.

According to the analysis, the relaxation time and the viscosity of the faster relaxation component stay constant within error in the WT and H3KQ cases. This may be because H3KQ would exert two opposing effects on the intra-array conformational dynamics of mobile arrays. One is to make the mobile arrays more dynamic due to intra-array DNA-histone dynamics facilitated by H3KQ. From the point of mobile arrays casting barriers to the motions of another body, this effect will

**Fig. 6 | Microrheology measurements of condensed droplets and the effects of histone chaperone Nap1 and H3KQ acetylation mimic. a** Experiments were set up based on optical tweezers to measure the dynamic modulus of nucleosome array droplets. A bead is sinusoidally oscillated at 7 different frequencies in a droplet. The chart shows the trap position oscillation (green) and the bead following the trap oscillation measured as the force oscillation (red). The bead oscillation is fit with a cosine wave (orange). The data are fit with a Burgers model with two Maxwell components. **b** Examples of dynamic modulus fitting and results for unmodified arrays (WT), WT arrays with Nap1 (Nap1), and arrays with the H3 tail acetylation mimic (H3KQ). **c** The fitting results are summarized for the two relaxation time components $\tau_0$ and $\tau_1$, the two viscosity components $\eta_0$ and $\eta_1$, and their sum $\eta_{tot}$, the two corresponding elastic modulus $\mu_0$ and $\mu_1$, and their sum $\mu_{tot}$. The sample size $n = 19$, 20, and 17, respectively for the WT, Nap1, and H3KQ cases. The sample size refers to the number of distinct droplets tested in each case. The

samples and the measurements were made on at least two different days. The values marked on the box plots represent mean ± standard deviation. The significances shown are from two-sided Student's $t$ test (*: $p \le 0.05$, **: $p \le 0.01$, ***: $p \le 0.001$, ****: $p \le 0.0001$, ns: not significant). The p-values for $\tau_0$ comparisons of WT with Nap1, WT with H3KQ, and Nap1 with H3KQ are 0.116, 0.659, and 0.231, respectively, those for $\tau_1$ are 0.000708, 0.000404, and 0.265, respectively, those for $\eta_0$ are 0.00105, 0.447, and 0.000137, respectively, those for $\eta_1$ are 0.0259, 0.0000929, and 0.000843, respectively, those for $\eta_{tot}$ are 0.0269, 0.0000928, and 0.000777, respectively, those for $\mu_0$ are 0.346, 0.319, and 0.255, respectively, those for $\mu_1$ are 0.588, 0.350, and 0.00204, respectively, and those for $\mu_{tot}$ are 0.262, 0.164, and 0.00149, respectively. Box plot elements are center line (median); box limits (upper and lower quartiles); whiskers (1.5x interquartile range). Source data are provided as a Source Data file.

make the bead motion more efficient, consequently resulting in a faster relaxation response and a lower viscosity. The other effect is from the perspective of the array motions themselves. The effect described above due to H3KQ will result in a larger hydrodynamic radius, thereby impeding the diffusion of mobile arrays. Such reduced diffusion would result in a slower relaxation response and a higher viscosity. Therefore, these two opposing effects will cancel each other, and the relaxation time and viscosity may stay apparently unchanged. This explanation can also account for the unchanged relaxation time observed in the presence of Nap1 although the viscosity is elevated likely due to a higher concentration of nucleosome arrays induced by Nap1 and the higher concentration of Nap1 in the condensates.

On the other hand, the slower relaxation component shows significantly shorter relaxation time and lower viscosity with Nap1 and H3KQ arrays than with WT arrays. These results suggest that the intra- and inter-array nucleosome dynamics in the structural scaffold take place on the order of a few hundred ms to a few seconds and that the network reconfiguration dynamics is significantly facilitated by Nap1 or H3KQ. Therefore, the faster FRAP recovery induced by Nap1 and H3KQ must be largely due to facilitated network reconfiguration dynamics of the structural scaffold in a condensate, which originates from eased intra- and inter-array nucleosome motions. This is further supported by the FRAP recovery times decreasing in the order WT > Nap1 > H3KQ, mirroring the decrease in the relaxation times and viscosities of the slower relaxation component. These results support that the viscous resistance of condensed chromatin can be lessened by histone chaperone and histone H3 tail acetylation, thereby facilitating the structural fluidity of intercalated nucleosome arrays at a chromatin level.

## Discussion

Nucleosome arrays are naturally prone to forming condensates via LLPS as they contain both unstructured positively charged protein chains and negatively charged DNA which are the main constituents of various LLPS systems[27,30,64]. Our results indicate that these condensed droplets reach nucleosome concentrations comparable to those found in the cell nucleus, serving as an excellent platform for investigating various chromatin components in a condensed phase. In condensed chromatin droplets, the dynamics of DNA-histone interactions and their regulations are not straightforward to model as nucleosomes are closely packed to each other while their motions are constrained by being linked in an array. One such example is spontaneous histone H2A-H2B exchange between nucleosomes. When nucleosomes freely diffuse in solution, they collide with each other and exchange H2A-H2B when the collision coincides with spontaneous partial unwrapping of nucleosomal DNA[65]. The timescale of spontaneous H2A-H2B exchange between freely diffusing nucleosomes at a few hundred μM concentration is on the order of a few seconds per nucleosome. However, it is hard to predict if such an exchange is feasible and significant in a condensed phase because nucleosome motions are constrained. Our

results show significantly facilitated diffusion dynamics of the nucleosomes labeled at H2A-H2B over those labeled at (H3-H4)$_2$ in the WT, the Nap1, and the H3ac cases (Fig. 4a). These results support spontaneous H2A-H2B exchange between nucleosomes within the structural scaffold of a condensate. This exchange would constitute a diffusion mechanism of H2A-H2B in condensed chromatin. The diffusion of H2A-H2B within condensed chromatin may provide an efficient way to remove their variants and post-translational modifications once their functions are fulfilled. Further investigation and quantitative analysis of the dynamics of modified H2A–H2B, including their epigenetic marks and variants, will be crucial for uncovering their biophysical mechanisms of gene regulation.

Acetylation of the lysine residues on the N-terminal tail of histone H4 have been associated mostly with inhibited inter-nucleosomal interactions[55,66]. The effect is due to the modulated H4 interactions with DNA and the acidic patch of the histone core across nucleosomes. It has been suggested that chromatin phase separation is induced by both short- and long-range inter-nucleosomal interactions between DNA and histone[67,68]. The short- and long-range interactions would be equivalent to intra- and inter-array interactions between nucleosomes in our system[69]. Both of these interactions would be inhibited by H4 tail acetylation. Our results indicate that such inhibition is sufficient to block chromatin phase separation of nucleosome arrays. The N-terminal tail of histone H3 also plays a role in inter-nucleosomal interactions, but to a lesser extent than that of H4, which is supported by droplet formation with H3KQ or H3 acetylated arrays and large aggregate formation with gH3 arrays. Therefore, our results suggest that H4-tail acetylation would exert a direct and strong impact on chromatin condensation and its accessibility. Deacetylations of H4 tail lysine residues have been implicated in chromatin condensation during mitosis and their acetylation is inversely correlated with inhibition of chromatin condensation in prophase[70,71].

Acetylation of the lysine residues on the N-terminal tail of histone H3 has been coupled to destabilized nucleosomes and elevated DNA accessibility within nucleosomes[31,32,54,55]. The effects are largely due to their weakened interactions with DNA. Such destabilization of the nucleosome structure is not to the extent where nucleosomes will disassemble spontaneously. Rather, it eases mechanical unwrapping of the outer region of nucleosomes, enhances thermal and salt-induced disassembly, and facilitates transcription by inhibiting an entry pause to the nucleosome[35,36,53]. These changes are correlated with making nucleosomes more flexible on the timescales of these events which are much longer than that of the spontaneous breathing motions of the nucleosome at milliseconds[60]. Such elevated flexibility would be observable as more dynamic motions of nucleosomes at both the nucleosome and the array levels. Our FRAP and microrheology results provide direct support to these changes by showing enhanced diffusion and faster relaxation on a few hundred ms to a few tens of seconds timescale. Our results also revealed that the facilitated motions and fluidity of nucleosomes and nucleosome arrays induced by H3KQ and

H3 tail acetylation are not sufficient to inhibit chromatin phase separation. Therefore, H3 tail acetylation would mainly function to enhance DNA-histone dynamics and protein diffusion within condensed chromatin.

Unlike gH4 arrays, gH3 arrays result in large-scale gel-like aggregates. As both tails are important in stabilizing DNA-histone interactions within and across nucleosomes, we suggest that these aggregates are seeded by disassembled nucleosomes upon their initial condensation. Nucleosome core particles containing gH3 or gH4 have nearly unchanged structures compared with canonical nucleosomes[72,73]. However, their dynamics and structural stabilities should be altered, as observed in our restriction enzyme digestion assay (Supplementary Fig. 10), which may result in the disassembly of nucleosomes upon closer and sustained non-canonical DNA-histone interactions in condensates. Histone chaperone Nap1 mediates these non-canonical DNA-histone interactions to facilitate nucleosome reassembly[43]. Our result showing Nap1-induced dissolution of gel-like aggregates strongly suggests that the aggregates are permeable gel-like and that they are formed by random non-canonical interactions between DNA and histone. Our restriction enzyme digestion assay also supports this mechanism (Supplementary Fig. 10). All in all, our results clearly show that histone H3 and H4 tail residues other than the lysine residues must play critical roles in stabilizing nucleosomes in condensed chromatin.

Faster FRAP recovery with Nap1 further confirms its role in dynamically mediating DNA-histone interactions. Such mediation has been reported from the thermodynamic perspectives[43,74]. However, it remained unknown how fast Nap1 mediates DNA-histone interactions to induce an observable change at a macroscopic level. Our FRAP results indicate that Nap1 facilitates DNA-histone dynamics within and across intact nucleosomes on the order of a few to a few tens of seconds, which is the diffusion timescales of nucleosomes in our chromatin condensates. This is a surprising point of Nap1's function in enhancing DNA-histone dynamics on this short timescale. As FRAP measurements report diffusion within and of the mobile fraction of the condensed droplet, they may not directly capture the properties of the immobile fraction which is important in chromatin condensation in vivo. The existence of the mobile and relatively immobile fractions are confirmed with the ~50 to ~60 % FRAP recovery and the STORM images. To this end, we investigated if the effect of Nap1 on enhancing DNA-histone dynamics applies to the relatively immobile structural scaffold with microrheology measurements. Unlike FRAP, these measurements report the properties of both the mobile and the immobile fractions of a droplet. Our results show that Nap1 enhances DNA-histone dynamics during network reconfiguration in the relatively immobile fraction on a relaxation timescale of a few hundred ms to a few seconds. This timescale is relevant to the diffusion and binding of various enzymes in chromatin such as RNA polymerase II and chromatin remodelers, and their actions in reading and altering chromatin[75]. Therefore, Nap1 likely promotes a dynamic yet condensed chromatin environment that facilitates nucleosome turnover and regulatory protein access. This property may underlie Nap1's physiological role in maintaining chromatin plasticity during processes such as transcription, DNA replication, and chromatin remodeling.

Our microrheology measurements revealed two relaxation components. The faster relaxation component is likely due to intra-array conformational dynamics that would be relevant to short-range internucleosomal interactions in condensed chromatin, while the slower relaxation component would be due to long-range inter-nucleosomal interactions. Therefore, the slower component would likely mimic a relatively immobile structural scaffold of condensed chromatin formed with long nucleosome arrays in vivo. Moreover, the markedly elevated chromatin fluidity observed with Nap1 addition and H3KQ modification are due to the facilitated reconfiguration of the structural scaffold, suggesting broader relevance to the regulation of condensed chromatin structure and dynamics.

## Methods

### DNA purification

The plasmid containing 12 repeats of the Widom 601 sequence with a 25 bp linker length, pWM+12×601_172NRL, was a generous gift from the Rosen lab. The 12×601 array DNA was prepared largely as previously described[27]. Briefly, we transformed the plasmid into Top 10 Competent Cells (Thermo Fisher Scientific, Waltham, MA) and plated it on LB agar plates supplemented with 100 µg/mL carbenicillin for growth overnight. After inoculating with a small-scale preculture from a single colony, 6L of LB culture with 100µg/mL carbenicillin was shaker-incubated overnight at 37°C. The culture was harvested by centrifugation, and the plasmid was purified with the Plasmid Maxi kit (Qiagen, Hilden, Germany). The plasmid was digested with restriction enzyme EcoRV (New England Biolabs, Ipswich, MA), and then the 12×601 array DNA was purified through size exclusion chromatography (SEC) with a Sephacryl S-500 HR (Cytiva, Marlborough MA) column (size = 26 mm inner diameter, height = ~90 cm, bed volume = ~478 mL) as shown in Supplementary Fig. 1. The purification step also generated competitor DNA fragments (~300 bp) used for later nucleosome array preparation (Supplementary Fig. 1). For SEC, the column was pre-equilibrated with two bed volumes of a buffer containing 10 mM Tris-HCl (pH 7.5), 5 mM NaCl, and 1 mM EDTA at a flow rate of 0.7 mL/min, followed by sample loading (~3 mL volume) at a rate of 0.9 mL/min. The sample was then eluted with the same buffer (10 mM Tris-HCl (pH 7.5), 5 mM NaCl, and 1 mM EDTA) at a rate of 0.7 mL/min. The elution was collected into 1.0 mL x 384 fractions.

To attach a biotin molecule at one end of the DNA, a plasmid containing 12 repeats of the Widom 601 sequence with a 25 bp linker length[76] (a gift from Dr. Sergei Grigoryev, the Pennsylvania State University College of Medicine) was digested with BamHI, HindIII, HaeII, and DraI (New England Biolabs) to leave the nucleosome array DNA with a 4-nucleotide overhang at both ends. The HaeII and DraI enzymes were to digest the plasmid backbone into shorter fragments. The digested plasmid was purified using the same protocols described above. Two short adapter DNA fragments (Integrated DNA Technologies, Coralville, IA) were ligated to the sticky ends of the purified nucleosome array DNA with T4 DNA ligase (New England Biolabs) at 16 °C for 14 h followed by inactivation at 65 °C for 20 min. The sequences of the adapter DNA fragments are phos-GATCCAGTACCTAGCATT-biotin and CTAGGTACTG for the BamHI end, and phos-AGCTTAAGCTGAGT and ACTCAGCTTA for the HindIII end, where phos- indicates 5' phosphorylation. The resulting DNA has one end labeled with a biotin molecule and was purified with a PCR clean-up kit (Qiagen). The purity was confirmed with a 1% agarose gel which showed a single band at the appropriate molecular weight.

### Nucleosome array assembly and purification

Wild-type human histones and other mutants (tailless histones, acetylation mimics, and histone octamers containing H4E63C/H3C96S C110A or H2BT115C/H3C96S C110A triple-mutations for fluorophore labeling) were purchased from the Histone Source (Colorado State University). For histone fluorophore-labeling, histones were incubated with a 10x molar excess of maleimide-functionalized Alexa Fluor 647 (Thermo Fisher Scientific) in a 10 mM Tris-HCl buffer (pH 7.5) for 2 h at room temperature, followed by dialysis to remove unreacted fluorophores. As previously described, 12-mer nucleosome arrays were reconstituted by dialyzing a mixture of histones and nucleosomal DNA at a fixed ratio in the presence of competitor DNA fragments in a dialysis device (Slide-A-Lyzer MINI Dialysis Device, 3.5 K MWCO, Thermo Fisher Scientific) against 1x TE (10 mM Tris-HCl, 1 mM EDTA, pH 8.0) buffer and stepwise decreasing salt concentrations of 2000, 1500, 1000, 750, 500, and 5 mM NaCl. The ratio of DNA:histone was

determined by titrating histone to DNA to ensure saturation of histone to nucleosomal DNA (Supplementary Fig. 2). We selected the ratio where the competitor DNA starts absorbing histone detectably on a gel and the nucleosome array band stops shifting upward at a higher DNA:histone ratio. The selected ratios are 1:1.4, 1:1.4, 1:1.4, 1:1.2, 1:1.2 for WT, gH3, gH4, H3KQ, and H4KQ arrays. It should be noted that the ratio depends on the freshness of a histone octamer sample as histone even in the octamer format is prone to precipitation over time. The assembled arrays were fractionated via size exclusion chromatography (Sephacryl S500-HR, Cytiva) (Supplementary Fig. 3). The protocols are the same as those used in DNA purification except for a slower elution rate of 0.3 mL/min and a smaller fraction size of 0.6 mL to increase the separation resolution, resulting in 384 fractions of a 230.4 mL total volume eluted over ~13 h. The quality of each fraction was assessed by TEM imaging (for arrays with no modifications, Supplementary Fig. 4) and analytical ultracentrifugation (AUC, Supplementary Fig. 3). See further details on the AUC and TEM methods in Supplementary Figs. 3 and 4. The fraction with the highest purity was selected for experiments according to the AUC results. The quality of the selected fraction was further confirmed with restriction enzyme digestion and SDS-PAGE. The Widom 601 sequence contains a BstUI restriction site at the 69th–73rd nucleotides (nt) from the nucleosome entry. This site becomes accessible for digestion only when the nucleosomal DNA unwraps due to disassembly or unsaturation during assembly. A 0.2 μg aliquot of the selected fraction of arrays was incubated with 1 unit of BstUI for 1 h at 37 °C in a 50 μL (final reaction volume) buffer containing 50 mM KOAc, 20 mM Tris-OAc (pH 7.9), 10 mM MgOAc, and 100 μg/mL albumin. The digested arrays were analyzed on a 1.5% agarose gel (Supplementary Fig. 4c). For further confirmation of a constant level of histone loading to the array samples, an SDS-PAGE analysis of the array samples was carried out. The result confirms approximately constant amounts of histones loaded on the array samples (Supplementary Fig. 4d).

## Nap1 purification

6xHis-Yeast nucleosome assembly protein 1 (Nap1) was expressed in *E. coli* and purified with Ni-NTA beads (Thermo Fisher Scientific) as reported in a previous publication[61]. Briefly, yeast Nap1 was expressed in *E. coli* cells with an N-terminal His-tag. The protein is purified from cell lysates via Ni-agarose affinity chromatography and further with a Mono-Q column, and the fractions were analyzed by SDS-PAGE to confirm the molecular weight.

## Purification of Piccolo NuA4 and Ada2/Ada3/Gcn5 HAT complexes

Full length Esa1, Yng2(1–218), and hexahistidine-tagged Epl1(51–380) subunits of yeast Piccolo NuA4 were coexpressed using the pST44 polycistronic expression vector in BL21(DE3)pLysS cells as described previously[77,78]. The Piccolo NuA4 complex was purified by Talon (Clontech) cobalt metal affinity chromatography, followed by removal of the hexahistidine tag by TEV protease digestion and subsequent SourceQ (Cytiva) anion-exchange chromatography.

The yeast Ada2/Ada3/Gcn5 complex containing full length Ada2 residues, C-terminally hexahistidine tagged Ada3(187-702) and full length Gcn5 was coexpressed using the pST44 polycistronic expression vector[78]. Protein expression in BL21(DE3)pLysS host strains was induced with 0.2 mM IPTG at 18 °C. The Ada2/Ada3/Gcn5 complex was purified from cell lysates by Talon (Clontech) cobalt metal affinity chromatography, followed by SourceQ (Cytiva) cation-exchange chromatography.

## Phase separation procedure and bright field imaging

Nucleosome arrays were stored in the final dialysis buffer (5 mM NaCl, 1X TE, 1 mM Dithiothreitol). The addition of the phase separation buffer containing 200–300 mM NaCl resulted in the final mixture composition of 150 mM NaCl, 25 mM Tris-HCl (pH 7.5), and 200 nM nucleosome arrays. After 30 min of incubation, the mixture was transferred to fluidic channels constructed on a microscope slide surface, which had been passivated with methoxy poly-ethyleneglycol (mPEG, 2 kDa molecular weight, Laysan Bio, Arab, AL) and bovine serum albumin (BSA, Millipore Sigma). Detailed methods of surface treatment and fluidic channel construction can be found elsewhere[79,80]. Briefly, pre-drilled microscope quartz slides were purchased from G. Finkenbeiner Inc. (Boston, MA). Coverslips and quartz slides were thoroughly cleaned following published procedures[80], including soapy water rubbing, KOH etching, and Nochromix® sulfuric acid solution treatment. Then, the surfaces of the cleaned coverslips and slides were coated with biotin-PEG-silane (MW 3400, Laysan Bio). The microscope fluidic channels were constructed between a coverslip and a slide with 6 thin strips of double-sided tape of 0.1 mm thickness. The tape strips were spaced at a ~1.5 mm interval to create 5 fluidic channels sandwiched between the coverslip and the slide. A 50 μL aliquot of a 100 mg/mL bovine serum albumin solution was injected into a fluidic channel and incubated for 15 min before transferring the sample.

Bright-field microscope images were taken on a Nikon Eclipse TE2000 inverted microscope with a Nikon Plan Apo 60x water-immersion objective. The pictures were recorded with an IAI CVM50 Industrial CCD Camera (147×114 μm², 752 ×582 pixels with a pixel size of 196 nm).

## Confocal imaging and FRAP measurements

Confocal fluorescence images were captured on a Zeiss LSM 880 confocal laser scanning microscope equipped with a 63×1.4 NA objective lens and 34-channel NLO array detectors. Fluorescently labeled arrays were mixed with unlabeled arrays and biotinylated arrays at a 2(fluorescent):7(non-fluorescent):1(biotinylated) molar ratio. Biotinylated arrays were mixed to immobilize droplets on a coverslip. A surface-passivated coverslip was used for imaging. The coverslip was coated with mPEG (2k Molecular weight, Laysan Bio, Arab, AL) and biotin-PEG (5k Molecular weight, Laysan Bio, Arab, AL). Detailed surface preparation protocols can be found elsewhere[35,36,53]. Streptavidin (0.2 mg/mL, Thermo Fisher Scientific) was first incubated on a coverslip and rinsed out with the phase separation buffer before droplets were loaded. Fluorescence Recovery After Photobleaching (FRAP) was achieved with the built-in bleaching function in the ZEN software package provided by the microscope manufacturer (Zeiss, Germany). A normalized fluorescence recovery trajectory was fit to an exponential function $FRAP(t) = A(1 - e^{-\frac{t}{\tau}})$ where τ is the recovery time and A is the recovery fraction.

## Microrheology measurements with optical traps

A C-trap optical tweezers instrument (Lumicks, Netherlands) was used to probe the viscoelasticity of a droplet by oscillating an optically trapped polystyrene bead (monodisperse analytical standard, 1 μm diameter, Supelco® via Millipore Sigma) inside. While the bead was oscillating, the trap position (i.e., bead position) and the force feedback were recorded. The temperature of the stage and the sample chamber were maintained at 25 °C during the measurements. The analysis followed previous publications[62,81]. Briefly, the trajectories of the bead position (X) and the force feedback (F) were fit to the following two equations to extract the elastic and viscous moduli:

$$X = X_o \cos(\omega t + \varphi) + A \quad (1)$$

$$F = F_o \cos(\omega t + \varphi + \Delta) + B \quad (2)$$

where $\omega$ is the angular frequency of bead oscillation, $X_o$ and $F_o$ are the amplitudes of the cosine waves representing the bead position and the

force feedback, respectively, $\varphi$ is a phase offset of the bead position oscillation, $\Delta$ is the phase lag between the bead position and the force feedback, and A and B are the amplitude offsets for the bead position and the force feedback, respectively. Once the values of $\omega, \varphi, X_o, F_o,$ and $\Delta$ were determined from the fitting, the following equations were used to compute the elastic (G′) and viscous (G″) moduli at the frequency ($\omega$). The force constant of the optical trap (k) was acquired prior to each experiment:

$$\Gamma = \frac{F_o}{kX_o} \tag{3}$$

$$G' = \frac{F_o}{6\pi a X_o} \frac{\cos\Delta - \Gamma}{(\cos\Delta - \Gamma)^2 + \sin^2\Delta} \tag{4}$$

$$G'' = \frac{F_o}{6\pi a X_o} \frac{\sin\Delta}{(\cos\Delta - \Gamma)^2 + \sin^2\Delta} \tag{5}$$

The obtained moduli values at 7 different frequencies were fit to a Burgers material model with two Maxwell components where the two relaxation components are linearly combined. The equations used for fitting are as follows. The values of $\tau$ and $\eta$ were obtained from the fitting:

$$G(\omega) = G'(\omega) + iG''(\omega) \tag{6}$$

$$G'(\omega) = \frac{\omega^2 \tau_0 \eta_0}{1 + (\omega\tau_0)^2} + \frac{\omega^2 \tau_1 \eta_1}{1 + (\omega\tau_1)^2} \tag{7}$$

$$G''(\omega) = \frac{\omega\eta_0}{1 + (\omega\tau_0)^2} + \frac{\omega\eta_1}{1 + (\omega\tau_1)^2} \tag{8}$$

where $\tau$ and $\eta$ are the relaxation time and viscosity, respectively, and the subscripts 0 and 1 are the indices of the two components. The two linearly combined relaxation components are more explicit in the time domain modulus as follows:

$$G(t) = \frac{\eta_0}{\tau_0}\exp\left(-\frac{t}{\tau_0}\right) + \frac{\eta_1}{\tau_1}\exp\left(-\frac{t}{\tau_1}\right) \tag{9}$$

The elastic moduli for a sudden and sustained strain are the amplitudes of the two components as follows:

$$\mu_0 = \frac{\eta_0}{\tau_0} \; and \; \mu_1 = \frac{\eta_1}{\tau_1} \tag{10}$$

### Stochastic optical reconstruction microscopy (STORM)

Three-dimensional (3D) STORM imaging was performed using a custom-built total internal reflection fluorescence (TIRF) microscopy based on a Nikon Eclipse Ti2-U inverted microscope body. A 642-nm laser (MPB Communications, 2RU-VFL-P-2000-642-B1R) was used for excitation of Alexa Fluor 647, while a 405-nm laser (Coherent, OBIS 405 nm LX) was used to reactivate fluorophores from the dark state during imaging. Both lasers were introduced through the microscope ́s back port and directed into a high numerical aperture (NA = 1.45) 100x oil-immersion objective via a multiband dichroic mirror (ZT405/488/561/640rpcv2, Chroma). Fluorescence emission was filtered using a quad-band emission filter (ZET405/488/561/640m-TRFv2, Chroma) and detected by an EMCCD camera (iXon Life 897, Andor Technology). A TIRF lens mounted on a translation stage allowed lateral displacement of the laser beams at the objective's back aperture, ensuring illumination at angles just below the critical angle of the glass–water

interface and selectively exciting fluorophores within ~2 μm of the coverslip surface. A cylindrical lens (f = 1000 mm) was introduced into the emission path to introduce axial astigmatism, causing the point spread function (PSF) of single-molecule fluorescence to appear elliptical when emitters were above or below the focal plane. This PSF ellipticity was used to determine the z-position of each molecule, enabling 3D STORM reconstruction[82]. The astigmatic single-molecule images were fitted with an elliptical Gaussian function to extract the centroid in the x–y plane, and the z-position was computed based on the ellipticity of the PSF, which systematically varies with axial displacement from the focal plane.

For STORM imaging, the chromatin condensate samples were mounted in an imaging buffer containing 25 mM Tris-HCl (pH 7.5), 150 mM NaCl, 5% (w/v) glucose, 10 mM cysteamine (MEA), 0.8 mg/mL glucose oxidase, and 40 μg/mL catalase. The condensates were formed with a mixture of 1:500 fluorescently labeled:unlabeled nucleosome arrays. The fluorophore was labeled at H4 E63C. During imaging, continuous 642-nm illumination (~2 kW/cm$^2$) was used to excite Alexa Fluor 647 molecules and switch them into a dark state, and continuous 405-nm illumination (0-1 W/cm$^2$) was applied to stochastically reactivate fluorophores to the emissive state. The laser power was adjusted to ensure that only a sparse, optically resolvable subset of fluorophores were active at any given time. Movies consisting of 40,000 image frames were acquired at 100 Hz with a pixel size of 160 nm. Each set of 1000 consecutive images, corresponding to a 10 s observation period, was processed to generate a 10 s Total STORM image. The full imaging duration of 400 s produced 40 of such 10 s STORM images. Within each 10 s STORM image, we identified clusters by grouping 5 or more histone H4 localizations within 110 nm, which is approximately the contour length of a 12-mer nucleosome array determined with TEM images. Retaining the identified clusters and removing other H4 localizations in a 10 s STORM image result in a 10 s Clusters STORM image.

### Reporting summary

Further information on research design is available in the Nature Portfolio Reporting Summary linked to this article.

## Data availability

The authors declare that the data supporting the findings of this study are available within the paper and its Supplementary Information files. Should any data be needed in another format they are available from the corresponding author upon reasonable request. Source data are provided with this paper.

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

## Acknowledgements

This research was supported by NIH grants R35GM148208 to T.L., R35GM142973 to R.Z., and R35GM127034 to S.T. The analytical ultracentrifuge and the C-Trap instrument used for this research were funded by NIH (S10OD032215 and S10OD038190).

## Author contributions

T.-H.L. conceptualized, initiated, and supervised the research. J.G., H.L., S. T., R.Z., and T.-H.L. contributed to the design of experiments. J.G. and H.L. performed experiments and data analyses. J.G., H.L., S.T., R.Z., and T.-H.L. contributed to writing and proofreading the manuscript.

## Competing interests

The authors declare no competing interests.
