## [Transparent Peer Review file · Nature Communications]

Roles of histone chaperone Nap1 and histone acetylation in regulating phase-separation of nucleosome arrays

Corresponding Author: Dr Tae-Hee Lee

Version 0:

Reviewer comments:

Reviewer #1

(Remarks to the Author)

Gao et al. describe new measurements of the biophysical properties of in vitro assembled chromatin condensates as proxy's of cellular chromatin. They assemble 12mer nucleosome arrays using intact histones and those that lack either the H3 or H4 tail or feature extensively mutated versions thereof. As expected, they find that the H4 tail is instrumental in chromatin liquid liquid phase separation (LLPS), whereas mutations of Lys residues in the H3 tail are tolerated and lead to chromatin phases with increased dynamics as determined by FRAP and microrheology. The histone chaperone NAP1 similarly increased chromatin dynamics. Furthermore, they identify microscopically heterogeneous assemblies that contain relatively immobile clusters (observed by STORM) as well as highly mobile areas. The work advances our understanding of chromatin dynamics in vitro although more detail and context are needed to untangle the molecular factors that govern these dynamics. The authors should therefore consider the following points before publication in Nature Comms can be recommended:

- The authors employ K->Q mutations and incubation with an acetyltransferase complex to probe the role of H3 acetylation in LLPS. However, K->Q mutations (although commonly employed in the field) are poor mimics of acetyllysine and the authors provide no evidence that incubation with Ada2/Ada3/Gcn5 leads to significant H3 acetylation. The authors should demonstrate H3 acetylation in condensates by Ada2/Ada3/Gcn5 and consider repeating the FRAP assays upon acetylation to test whether HAT activity can enhance dynamics as they surmise based on the extensive K->Q mutations.

- Addition of Nap1 to chromatin condensates led to a surprising increase in H3/H4 concentration. This measurement should be repeated with di-labelled arrays to shed light on the contents of the liquid phase.

- Along the same lines, are stoichiometric amounts of Nap1 required for altered dynamics? Is this ratio biologically relevant? If not, catalytic amounts of Nap1 should be tested to explore whether their presence increases dynamics by helping to shuttle H2A/HAB between nucleosomes, rather than disassembling them.

- The authors state that "the entirety of the droplet interior is full of relatively static nucleosomes...". Can this statement be quantified, e.g. by comparing the radial distribution of mobile and immobile nucleosome fractions? Is this observation unique to nucleosomes or mirror other LLPS systems?

- The raw data underpinning Figure 4 should be appended to the SI

- Please indicate how many replicates were conducted for the measurements in Figures 2, 3 and 5 and provide the additional data in the SI

- (minor) The authors may want to consider relegating Fig 3 to the SI because it addresses an artefact originating from the gH3 construct.

Reviewer #2

(Remarks to the Author)

This article by Gao et al. reports a study on the LLPS behavior of a 12-mer nucleosome array and its modulation by H3/H4 tail acetylation and Nap1. Using FRAP, STORM, and optical tweezers, the authors identified two components (immobile and

mobile arrays) within nucleosome droplets. While this study provides interesting insights into how H3, H4, and Nap1 modulate nucleosome droplet mobility, it lacks *in vivo* study to demonstrate the biological relevance of this LLPS modulation. Additionally, some claims—such as the assertion that H4 tail lysine residues are the main drivers of chromatin LLPS—are underdeveloped and lack sufficient molecular-level evidence or discussion. For instance, the role of specific residues in driving LLPS, and why no other residues are involved, requires further exploration. Overall, I feel this article is better suited for a more specialized journal.

Below are additional detailed comments:

1. The authors used FPLC to purify the final nucleosome array products. However, in Fig. S3, only a small fraction of the 12-mer nucleosome array was selected, while a larger fraction of the reconstitution products (as shown in Fig. S2) appears to correspond to the array. Why was only a small fraction chosen (Fig. S3, black box)? What are the species on the right side of the same elution peak? Why were these species not shown in the gel?
2. In Fig. 2, gH3 and H3KQ promote nucleosome association, and gH4 also exhibits such an effect, albeit to a lesser extent. This suggests that both H3 and H4 tails contribute to nucleosome array association. It may be that droplets were seen for H3KQ and aggregates were observed for gH3 at this specific concentration and salt strength (and, why 200 mM NaCl instead of 150 mM as in Fig. 1). To conclude that H4 tail lysine residues are the main drivers of chromatin LLPS while H3 tails are not, multiple parameters such as array concentration and salt concentration should be screened to generate phase separation diagrams. Phase diagrams should also be considered for Fig. 2b and c. Additionally, aggregation formation in Figs. 2 and 3 should be analyzed more quantitatively (e.g., measuring array concentration in the supernatant after centrifugation) rather than relying solely on microscopy images.
3. In Fig. 3, if Nap1 disassembles gH3 nucleosome array aggregates, it should also affect nucleosome array LLPS. However, this is not observed for WT + Nap1. How can this be explained? The disassembly of nucleosome arrays should be determined using techniques like analytical ultracentrifugation.
4. The observation of immobile nucleosome arrays were done by 10-second period of STROM imaging. Could the author comment in the manuscript why 10-second is chosen here?
5. In methods, experiment details of TEM and AUC are not provided. Some other experiment details are also missing and only references are given. Those details need to be provided.

Reviewer #3

(Remarks to the Author)

The manuscript by Gao et al characterizes the role of the histone chaperone Nap1 as well as histone H3 and H4 acetylations in regulation of liquid-liquid phase separation (LLPS) of chromatin employing a 12-mer 601 Widom nucleosome array. They identify the histone H4 tail as a primary driver of chromatin liquid-liquid phase separation, with a less significant but a contributing role for histone H3 tails. They present and discuss the contribution of H3 and H4 tail acetylation in mediating the chromatin phase separation. They further show that the histone chaperone Nap1 plays a role in regulating the condensate in resolving aggregates and enhancing fluidity of the condensates. They also demonstrate that there is a scaffold of relatively static chromatin elements within the chromatin droplets. Finally, they show that chromatin droplets with acetylation mimic and Nap1 has a lower viscosity. The paper adds important additional details to the understanding of the mechanisms of chromatin LLPS and should be published after major revision.

Comments:

- Generally linguistic review would help to improve the text. See some examples below
- The title should use the more well-defined term “nucleosome array” not “chromatin array”. Same in the abstract where this is used once. In the rest of the text they correctly use “nucleosome array”.
- “Chromatin condensation is dynamically regulated throughout the cell cycle and plays key roles in modulating gene accessibility at the highest physical level in a cell.”: Suggests rephrasing “highest physical level”, it is unclear what this means.
- The fundamental “gene packing” unit in chromatin is the nucleosome, which is ~150 bp DNA wrapped around an octameric histone core containing two H2A-H2B dimers and one (H3-H4)₂ tetramer. “DNA packaging” is more accurate.
- “...undergo liquid-liquid phase separation (LLPS) to form liquid-like droplet condensates 27,28.” Relevant missing reference: <https://doi.org/10.3390/cells11193145>
- “spherical, suggesting that the droplets begin in a liquid-like state and age into a gel-like state 29.” The findings of this paper are largely disproved by reference 27 that proposed that the observations may be due to sample preparation issues. Hence, conclusions should be taken with a grain of salt.
- “acetylations on H3 or H4 have been implicated in weakened intra- and inter-nucleosomal DNA-histone interactions 30–35” A key reference missing: <https://doi.org/10.1093/nar/gkq900> Another key reference missing here or when tailless construct results are discussed is:
doi: 10.1016/s0022-2836(03)00025-1
- Methods: There are no details on size exclusion chromatography of 12-mer DNA and assembled arrays and analytical ultracentrifugation. It is strongly suggested adding these details.
- Figure S1. An agarose gel image of fractions showing the separation of the DNA would be a good addition.
- Figure S2. It would be ideal to indicate the ratio for each band in the gel image and also the determined optimal saturation ratio in the gel. Is the loading of chromatin the same for all gel panels? It appears that the amount of competitor DNA used is different for the different ratios unless this is an artifact due to different gel run times.
- Comment on size exclusion chromatography purification of assembled arrays: This is not a widely employed protocol

though preparative methods are used to separate free DNA and mononucleosome. An agarose gel for the SEC purified chromatin sample and SDS-gel verification of the presence of 4 histones is recommended. In S3, there are significant differences in the sedimentation rate of different arrays. Based on the findings of the manuscript one would expect H4KQ and gH4 to have the most open chromatin however this is not reflected in the AUC results. SDS analysis is suggested of the chromatin to ensure there was no dimer loss during the size exclusion chromatography of the chromatin, especially for the gH3 sample owing to its lower stability. As the authors have established a protocol for TEM imaging, alternative to SDS/gel an analysis of the mutant and globular arrays that have gone through SEC would also suffice.

- “Among these four samples, only the H3KQ arrays result in chromatin droplets (Fig. 2a).”: A comment on comparison with wild type would be informative to the reader.

- Page 11: “Upon a closer look at the droplets, we observed crumbling from the inside, loss of the round outer fringe, and shrinkage of the droplet size within 10 – 15 minutes”: An image would be ideal, please add in supplementary.

- Page 11: “We hypothesize that the aggregation is caused by nucleosome disassembly upon nucleosome destabilization due to the lack of the H3 tails.” A TEM imaging of the arrays might show evidence for this, in form of aggregates or disassembled nucleosomes.

- “These results validate our hypothesis of disassembled nucleosomes and random histone-DNA mixture in condensed chromatin containing tailless H3.”: The disassembly of nucleosomes was not confirmed with direct/indirect observation; hence the above statement is not accurate.

Comment: Nucleosomes are sensitive to cations, and this was exploited in obtaining nucleosome crystals. The crystallization experiments often resulted in precipitation/aggregates that probably might not have been due to disassembly. The aggregates observed might possibly be trapped nucleosome-nucleosome interaction that might not be the most energetically favored interaction. The Nap1 probably might just be ‘releasing the trapped nucleosomes’ and facilitating the most energetically favorable nucleosome-nucleosome interactions. However, if the authors can validate the disassembly, it would be very nice.

- The results in the later part though reasonable and logical, would be convincing if backed with a thorough analysis of the chromatin sample preparation.

- The preparation of droplets by addition of phase separation buffer:

The details of the procedure are not provided; hence the query is: is the concentration of salt in the buffer very high in order to induce dissociation in the gH3 and gH4 samples? Have the gH3 arrays been confirmed not to have dimer loss during SEC? 601 nucleosomes have relatively high stability, apart from reported breathing and dynamics in literature, the nucleosome requires high molarity salt to induce dissociation.

Version 1:

Reviewer comments:

Reviewer #1

(Remarks to the Author)

The authors' revisions have strengthened the manuscript and I recommend its publication in its current form.

Reviewer #2

(Remarks to the Author)

The authors have addressed most of my concerns. There are a few additional points to be considered before recommending the manuscript for publication in Nature Communications:

1. The extent of acetylation and probably also the precise acetylated sites of H3 catalyzed by Ada2/Ada3/Gcn5, and H4 by Piccolo NuA4 should be accessed with proper mass spectrometry techniques. This information is essential to well support the conclusions about the roles of different acetylation in the phase behaviors.

2. Typo in Fig 2 caption, “clearing showing a similar.....”

Reviewer #3

(Remarks to the Author)

The manuscript has significantly improved with the additions and amendments. The major conclusions of H4 being primary driver, role of Nap1, mobile and structural scaffold are a welcome addition to our understanding of chromatin phase separation.

Some minor comments:

Some review of certain phrase is still strongly recommended to improve readability and clarity of the ideas being presented. Few examples are listed below:

Abstract:

Suggest reviewing the following as it is difficult to comprehend:

“We also show that the condensed liquid-like droplets contain both a mobile fraction and a relatively immobile structural

scaffold and that histone chaperone Nap1 and histone H3 tail acetylation facilitate DNA-histone dynamics within the structural scaffold to lower the overall viscosity of the droplets.”

Introduction “The structure of chromatin is formed largely by arrays of genomic DNA, histone proteins, and various other chromatin-associated proteins.”

The structure of chromatin is still debated; “chromatin is composed of” is more appropriate.

74: Suggest to amend or refrain from claim of primacy. Ref 37 reports effects of H4 tail acetylation on chromatin folding and self-association, which is analogous to condensation behavior of chromatin arrays.

367 ‘at physiological nucleosome concentration’ might not be accurate. The nucleosome concentration in the experiments is 200 nM, though it reaches much higher concentration approaching physiological concentration within the droplets.

509-520 Some of the details are a better fit within methods

Nap1 increasing both nucleosome concentration and fluidity is fascinating. A comment on its physiological relevance would be good.

Reviewer #1 (Remarks to the Author)

Gao et al. describe new measurements of the biophysical properties of in vitro assembled chromatin condensates as proxy's of cellular chromatin. They assemble 12mer nucleosome arrays using intact histones and those that lack either the H3 or H4 tail or feature extensively mutated versions thereof. As expected, they find that the H4 tail is instrumental in chromatin liquid liquid phase separation (LLPS), whereas mutations of Lys residues in the H3 tail are tolerated and lead to chromatin phases with increased dynamics as determined by FRAP and microrheology. The histone chaperone Nap1 similarly increased chromatin dynamics. Furthermore, they identify microscopically heterogeneous assemblies that contain relatively immobile clusters (observed by STORM) as well as highly mobile areas. The work advances our understanding of chromatin dynamics in vitro although more detail and context are needed to untangle the molecular factors that govern these dynamics. The authors should therefore consider the following points before publication in Nature Comms can be recommended:

We appreciate the constructive comments from the reviewer. We addressed all the comments as detailed below.

- The authors employ K->Q mutations and incubation with an acetyltransferase complex to probe the role of H3 acetylation in LLPS. However, K->Q mutations (although commonly employed in the field) are poor mimics of acetyllysine and the authors provide no evidence that incubation with Ada2/Ada3/Gcn5 leads to significant H3 acetylation. The authors should demonstrate H3 acetylation in condensates by Ada2/Ada3/Gcn5 and consider repeating the FRAP assays upon acetylation to test whether HAT activity can enhance dynamics as they surmise based on the extensive K->Q mutations.

We are providing results of western blotting assays confirming array acetylation by Ada2/Ada3/Gcn5 in Fig. S10. We employed an antibody for acetylated H3K14, which has been previously utilized to probe the acetylation activity of this complex. This histone acetyltransferase (HAT) complex has the histone acetylation activity similar to that of the entire SAGA complex. It should be noted that the exact locations and extents of H3 acetylation by this HAT (and the entire SAGA complex) are yet to be clearly defined although some tail residues such as K9, K14, K18, and K23 have been reported to be acetylated by the SAGA complex with varying efficiencies. This uncertainty is the main reason why we choose to use the H3KQ mutant and test the effects of charge removal upon H3 tail lysine acetylation.

We repeated the FRAP assays with acetylated arrays by Ada2/Ada3/Gcn5. The results clearly demonstrate that the acetylated arrays by Ada2/Ada3/Gcn5 show a considerably faster FRAP recovery, confirming enhanced DNA-histone dynamics similarly to the K to Q mutations. Of note, the effect is not as strong with the acetylated arrays as with the H3KQ arrays. This may be due to incomplete acetylation by the enzyme in both the acetylation extent and the number/range of acetylated residues. Regardless, the difference is considerable and confirms the effect of acetylation by an enzyme on enhancing nucleosome dynamics in condensates.

These new results have been added to Fig. 4, Fig. S10, and the main text and the figure caption have been revised accordingly. The major addition is as follows (pages 14-15):

"To further confirm the effect, we employed histone acetyltransferase (HAT) Ada2/Ada3/Gcn5 to acetylate H3. The droplets formed with H3-acetylated (H3ac) arrays by this HAT complex display considerably faster FRAP recovery compared to the WT arrays with no changes in the recovery fraction (Fig. 4a). This HAT complex has a similar acetylation activity as the entire SAGA complex that acetylates mainly H3 tail lysine residues⁶⁶. Although the exact locations and the extent of acetylation by Ada2/Ada3/Gcn5 are yet to be clearly defined, some of the H3 tail lysine residues such as K9, K14, K18, and K23 have been reported to be acetylated by the entire SAGA complex with varying efficiencies^{67,68}. Our western blotting result also confirms acetylation at H3K14 as previously published (Fig. S10)⁶⁹. These results confirm that H3 acetylation considerably enhances DNA-histone dynamics in condensates."

- Addition of Nap1 to chromatin condensates led to a surprising increase in H3/H4 concentration. This measurement should be repeated with di-labelled arrays to shed light on the contents of the liquid phase.

We now added a new dataset to show that the concentration of Di-labeled arrays in droplets also increases in the presence of Nap1 (Fig. S16). The p-value for the difference is ≤ 0.0001 according to a two-sided student's t-test although the extent of increase is smaller than in the Tet-labeled arrays. Nevertheless, the increased concentration of Di-labeled nucleosomes is the same as the increased concentration of the Tet-labeled nucleosomes within error at the 1σ confidence interval (413 ± 24 vs 491 ± 56 μ M). In summary, the additional data confirms the increased nucleosome array concentration in droplets in the presence of Nap1. The added main text (page 16) is as following:

"We also found that the nucleosome concentration increases significantly upon the addition of Nap1 from 326 ± 26 (WT) to 491 ± 56 μM (Nap1). The Di-labeled arrays also show a concentration increase (371 ± 36 vs 413 ± 24 μM , see Fig. S16). Although the level of increase is smaller, the difference is significant according to the student's t-test (two-sided $p \leq 0.0001$, see Fig. S11). The increased concentration of Di-labeled nucleosomes is the same as the increased concentration of the Tet-labeled nucleosomes within error at the 1σ confidence interval (413 ± 24 vs 491 ± 56 μM)."

- Along the same lines, are stoichiometric amounts of Nap1 required for altered dynamics? Is this ratio biologically relevant? If not, catalytic amounts of Nap1 should be tested to explore whether their presence increases dynamics by helping to shuttle H2A/HAB between nucleosomes, rather than disassembling them.

Nap1 is abundant in cells at a few micromolar concentration although its precise concentration remains unknown. The stoichiometric amount of Nap1 to the nucleosome could vary from 1:2 to 1:8 depending on how many histone units per nucleosome Nap1 interacts with, although the minimum would be 1:2 as Nap1 forms a dimer and single Nap1 interacts with one histone molecule. In this revision, we tested Nucleosome:Nap1 ratios of 1:0.1, 1:2, 1:4, and 1:8 to find that a stoichiometric amount of Nap1 is required to induce a significant change in the FRAP-recovery time in condensates. A catalytic amount of 1:0.1 does not result in any noticeable change in the dynamics, suggesting that the enhanced DNA-histone dynamics is the result of Nap1 facilitating transient and repetitive nucleosome disassembly/reassembly in the H2A-H2B region rather than completely disassembling them. The observation that the enhancement is plateaued at 1:4 ratio also supports this mechanism as Nap1 interacts mainly with H2A-H2B unless the nucleosome is considerably disassembled. We also support this conclusion by showing that Nap1 is enriched in the condensed phase according to the results from an SDS-PAGE assay (Fig. S14). These results suggest that the Nap1's role in making nucleosomes more dynamic in condensed chromatin is localized where Nap1 directly interacts with nucleosomes. These results have been added to Fig. S13 and explanations have been added to the figure caption and the main text (page 15):

"To further support this mechanism, we monitored the FRAP recovery dynamics of Di-labeled nucleosome arrays at varying ratios of nucleosome:Nap1 at 1:0.1, 1:2, 1:4, and 1:8 (Fig. S13). The results show that a stoichiometric amount of Nap1 is required to induce a significant change in the FRAP-recovery time in condensates. A catalytic amount of Nap1 at 1:0.1 does not result in any noticeable change in the dynamics, suggesting that the enhanced FRAP recovery is the result of Nap1 facilitating transient and repetitive nucleosome disassembly/reassembly in the H2A-H2B region rather than completely disassembling them. The observation that the enhancement is plateaued at 1:4 nucleosome:Nap1 ratio also supports this mechanism as Nap1 interacts mainly with H2A-H2B unless the nucleosome is considerably disassembled⁴³. We also support this conclusion by showing that Nap1 is enriched in the condensed phase according to the results from an SDS-PAGE assay (Fig. S14). These results suggest that the Nap1's role in making nucleosomes more dynamic in condensed chromatin is localized where Nap1 directly interacts with nucleosomes."

- The authors state that "the entirety of the droplet interior is full of relatively static nucleosomes...". Can this statement be quantified, e.g. by comparing the radial distribution of mobile and immobile nucleosome fractions? Is this observation unique to nucleosomes or mirror other LLPS systems?

This type of analysis for LLPS systems has never been reported up to our knowledge. Therefore, we do not know if this is unique or not. But we suspect that systems transitioning into gel-like droplets from liquid-like ones upon aging would likely have this characteristic.

Our results show that the entire interior of a droplet is full with both mobile and immobile fractions. Their relative fractions may depend on many parameters, which is currently pursued as a separate project but is out of scope for this paper. For example, how the motions of nucleosomes/arrays get slowed down near the boundary depends a lot on the radial location and the conformation/charge distribution of the arrays, whose quantification is way beyond the scope of this paper. Instead, we show that the entire droplet interior is full with both mobile and immobile fractions with additional images of droplets of various sizes (Fig. S17), where both mobile and immobile fractions (on the 10-sec timescale) are evident. We are also adding a few snapshots from a movie to show that mixing of the contents of two droplets do not occur immediately upon fusion. Instead, when droplets fuse, their boundaries first fuse to form one droplet with two distinct interior phases, and then their contents mix in the next few seconds (Fig. S18).

We also added some of these explanations in the revision (page 18) as follows:

"Such an immobile fraction is evident in mixing the contents of two droplets during a fusion event (Fig. S18). During fusion, the boundaries of two droplets first fuse to form one elongated droplet with two distinct interior phases, and then their contents mix in the next few seconds."

- The raw data underpinning Figure 4 should be appended to the SI

We added a typical fitting result and the entire numerical data underpinning Fig. 4 to the SI (Fig. S11 and Table S1).

- Please indicate how many replicates were conducted for the measurements in Figures 2, 3 and 5 and provide the additional data in the SI

We added the information on the replicates in figure captions and some additional images (for Figs. 2, 3, and 5) to SI (Figs. S6, S9, and S17). All the observations were made multiple times reproducibly with two or more samples prepared on different dates.

- (minor) The authors may want to consider relegating Fig 3 to the SI because it addresses an artefact originating from the gH3 construct.

We believe that the results with gH3 arrays inform the readers of an important role for histone H3 tails in chromatin condensation, and as such, decided to leave Fig. 3 in the main text. Our results indicate that H3 tails inhibit random contacts among nucleosomes leading to aggregation, thereby helping proper liquid-like condensation of chromatin.

Reviewer #2 (Remarks to the Author):

This article by Gao et al. reports a study on the LLPS behavior of a 12-mer nucleosome array and its modulation by H3/H4 tail acetylation and Nap1. Using FRAP, STORM, and optical tweezers, the authors identified two components (immobile and mobile arrays) within nucleosome droplets. While this study provides interesting insights into how H3, H4, and Nap1 modulate nucleosome droplet mobility, it lacks in vivo study to demonstrate the biological relevance of this LLPS modulation. Additionally, some claims—such as the assertion that H4 tail lysine residues are the main drivers of chromatin LLPS—are underdeveloped and lack sufficient molecular-level evidence or discussion. For instance, the role of specific residues in driving LLPS, and why no other residues are involved, requires further exploration. Overall, I feel this article is better suited for a more specialized journal.

We appreciate the constructive comments from the reviewer. We addressed all the comments as detailed below.

Below are additional detailed comments:

1. The authors used FPLC to purify the final nucleosome array products. However, in Fig. S3, only a small fraction of the 12-mer nucleosome array was selected, while a larger fraction of the reconstitution products (as shown in Fig. S2) appears to correspond to the array. Why was only a small fraction chosen (Fig. S3, black box)? What are the species on the right side of the same elution peak? Why were these species not shown in the gel?

Our protocols for nucleosome array reconstitution are to saturate but not over- or under-saturate arrays with histone. These slightly under- and over-saturated arrays constitute respectively the later and the earlier fractions than the selected fractions (i.e., the right and the left sides to the black boxed fractions in Fig. S3a). These arrays are inseparable from the selected arrays on a gel as their molecular weight distribution varies gradually. No method is currently available to make 12-mer nucleosome arrays that are mono-dispersed in the number of nucleosomes per array. As such, reconstituted 12-mer nucleosome arrays have a distribution in the number of nucleosomes. Our selection of certain fractions from a size exclusion column is to ensure a narrow distribution of the array size (i.e., 12-mer per array on average) and a high level of sample homogeneity. We do not believe that this selection process is necessary, but nonetheless used narrowly selected arrays for higher sample homogeneity.

In this revision, we confirmed the sample homogeneity with a restriction enzyme digestion assay. In our array DNA, each nucleosomal DNA fragment (the Widom 601 sequence) contains a BstUI restriction site at the 69th-73rd nt from the entry site. This site would be accessible and digested by BstUI only if the site is empty (i.e., array under-saturated). We provide an agarose gel image (Fig. S4c) to show undetectable digestion, confirming a high level of saturation and homogeneity of our array samples.

We further confirmed the quality of assembled arrays with an SDS-PAGE assay, showing a constant level of histone loading to array samples (Fig. S4d). We added these additional results and explanations in the revision as follows (page 6):

“The quality of the selected fraction was further confirmed with restriction enzyme digestion and SDS-PAGE. The Widom 601 sequence contains a BstUI restriction site at the 69th – 73rd nucleotides from the nucleosome entry. This site becomes accessible for digestion only when the nucleosomal DNA unwraps due to disassembly or unsaturation during assembly. A 0.2 µg aliquot of the selected fraction of arrays was incubated with 1 unit of BstUI for 1 hour at 37 °C in a 50 µL (final reaction volume) buffer containing 50 mM KOAc, 20 mM Tris-OAc (pH 7.9), 10 mM MgOAc, 100 µg/mL albumin. The digested arrays were analyzed on a 1.5 % agarose gel (Fig. S4c). For further confirmation of a constant level of histone loading to the array samples, an SDS-PAGE analysis of the selected fractions was carried out. The result confirms approximately constant amounts of histones loaded on the array samples (Fig. S4d).”

2. In Fig. 2, gH3 and H3KQ promote nucleosome association, and gH4 also exhibits such an effect, albeit to a lesser extent. This suggests that both H3 and H4 tails contribute to nucleosome array association. It may be that droplets were seen for H3KQ and aggregates were observed for gH3 at this specific concentration and salt strength (and, why 200 mM NaCl instead of 150 mM as in Fig. 1). To conclude that H4 tail lysine residues are the main drivers of chromatin LLPS while H3 tails are not, multiple parameters such as array concentration and salt concentration should be screened to generate phase separation diagrams. Phase diagrams should also be considered for Fig. 2b and c. Additionally, aggregation formation in Figs. 2 and 3 should be analyzed more quantitatively (e.g., measuring array concentration in the supernatant after centrifugation) rather than relying solely on microscopy images.

We tested the extent of phase separation and droplet formation at varying concentrations of the arrays (25, 50, 100, and 200 nM) and NaCl (5, 50, 100, and 150 mM) utilizing UV-Vis absorbance measurements, and constructed phase diagrams of all cases shown in Fig. 2. Droplets or aggregates scatter light at wavelengths according to their sizes. The scattering extent in a range of wavelengths (350 – 840 nm) was measured as absorbances in a UV-Vis spectrometer. Non-zero detectable absorbance in this range confirms the formation of droplets or aggregates down to a few hundred nanometers in size. The abundances of the droplets and aggregates of various sizes measured as the integrated absorbance (350 – 840 nm) are presented as phase diagrams (Fig. 2bcd). The phase diagrams clearly show less efficient droplet formation with H3KQ arrays or WT arrays acetylated with Ada2/Ada3/Gcn5 than WT arrays at given array and NaCl concentrations, and no droplet formation with H4KQ arrays or WT arrays acetylated with Piccolo NuA4 under these conditions. We also carried out a UV-Vis assay to confirm aggregates vs droplet formation after centrifugation. We clearly show that aggregates completely sediment upon centrifugation while droplets form a bottom layer of denser nucleosome arrays upon centrifugation (Fig. S7). The new data and explanations have been added to Figs. 1 and 2 and the main text (pages 11-12):

“To survey the extents of droplet formation under a range of conditions, the UV-Vis absorbances of the arrays were measured at varying array concentrations (25, 50, 100 and 200 nM) and NaCl concentrations (5, 50, 100 and 150 mM). Droplets scatter light at wavelengths according to their sizes. The scattering extent was measured as a UV-Vis absorbance spectrum within the range of 350 – 840 nm. Non-zero detectable absorbances in this range confirm the formation of droplets down to a few hundred nanometers in size. A time series of changes in the UV-Vis spectrum during the first 30 minutes of droplet formation is shown in Fig. S5. As the changes are not monotonic increase or decrease at a specific wavelength in this range, we used the integrated UV-Vis absorbance (350 – 840 nm) after 30 minutes of droplet formation as a measure of the abundance of the droplets of various sizes. The measured droplet abundances are presented as a phase diagram (Fig. 1b).”

“The phase diagrams (Figs. 1b and 2b) clearly show less efficient droplet formation with H3KQ arrays than WT arrays at given array and NaCl concentrations, and no droplet formation with H4KQ arrays. The phase diagram of gH3 arrays was constructed in the same way as the other phase diagrams (i.e., integrated scattering strengths in the range of 350 – 840 nm). The small amount of aggregates formed with gH4 arrays is not detectable with a UV-Vis spectrometer. We also confirm the formation of aggregates that are distinct from droplets with UV-Vis measurements of the supernatant and the bottom fraction of the aggregate and droplet samples upon centrifugation (Fig. S7).”

“We acetylated unmodified (WT) arrays with Piccolo NuA4 and tested phase separation to observe no droplet formation under various conditions (see the phase diagram in Fig. 2c).”

“As expected, no changes were observed in droplet formation under various conditions, further validating the observation with the H3KQ acetylation mimic (Fig. 2d).”

3. In Fig. 3, if Nap1 disassembles gH3 nucleosome array aggregates, it should also affect nucleosome array LLPS. However, this is not observed for WT + Nap1. How can this be explained? The disassembly of nucleosome arrays should be determined using techniques like analytical ultracentrifugation.

Nap1 does not disassemble nucleosomes that are properly assembled, which is the case with the WT arrays. Nap1 can dismantle and reassemble nucleosomes by shuttling histones from/to nucleosomes only when the nucleosome has already been disassembled considerably. Nap1 may transiently interact with H2A-H2B and even slightly displace them within a nucleosome when DNA-H2A-H2B interactions are temporarily weakened.

When DNA and histones are mixed, they form aggregates due to their strong and random electrostatic interactions. Therefore, once array nucleosomes are at least partially unwrapped as in the gH3 arrays, the arrays would form large aggregates and precipitates due to the exposed histones that would interact with DNA randomly to neutralize their strong positive charges. We show that gH3 array nucleosomes are partially unwrapped near the nucleosome termini and that Nap1 facilitates re-wrapping of these nucleosomes. We did this by employing restriction enzymes HinfI and BsrBI (Fig. S10). The 601 nucleosome sequence contains a HinfI restriction site at the 5th – 10th nt from the entry and a BsrBI restriction site at the 23rd to 28th nt from the entry. Partially unwrapped nucleosomes at the nucleosome termini will make the HinfI site accessible and digested while more completely disassembled nucleosomes will make the BsrBI site accessible and digested. Our digestion assay shows that the gH3 arrays are digested by HinfI while they are not digested by BsrBI, supporting partial disassembly in the aggregates near the entry/exit region of the nucleosomes. Upon Nap1 treatment to dissolve most of the aggregates, we see a much lower level of digestion by HinfI, confirming that some of the partially unwrapped nucleosomes are re-wrapped. The following and Fig. S10 have been added to the revision (page 14):

“To examine the extent of nucleosome disassembly in gH3 arrays, we employed restriction digestion assays with two restriction enzymes HinfI and BsrBI. The 601 nucleosome sequence contains a HinfI restriction site at the 5th – 10th nt from the entry and a BsrBI restriction site at the 23rd to 28th nt from the entry. Partially unwrapped nucleosomes at the nucleosome termini will make the HinfI site accessible and digested by the enzyme while more completely disassembled nucleosomes will make the BsrBI site accessible and digested. Our digestion assay shows that the gH3 arrays in the aggregates are digested by HinfI while they are not digested by BsrBI, supporting partial disassembly in the aggregates near the entry/exit region of the nucleosomes (Fig. S6). Upon Nap1 treatment to dissolve most of the aggregates, we see a much lower level of digestion by HinfI, confirming that some of the partially unwrapped nucleosomes are re-wrapped.”

4. The observation of immobile nucleosome arrays were done by 10-second period of STROM imaging. Could the author comment in the manuscript why 10-second is chosen here?

The 10-sec time scale was chosen to be similar to the droplet fusion time scale (typically <10 sec for droplets of a few micrometer diameter). This time scale ensures that we can observe both relatively mobile and immobile nucleosomes in droplets. This information (“This time scale is similar to the droplet fusion time scale (typically <10 seconds for droplets of a few μm diameter) so that we can have both relatively mobile and immobile fractions.”) has been added to the revision (page 17).

5. In methods, experiment details of TEM and AUC are not provided. Some other experiment details are also missing and only references are given. Those details need to be provided.

We added further details of TEM and AUC in the methods section and in the captions of Figs. S3 and S4. We also added more details of DNA purification, nucleosome assembly and purification, restriction enzyme digestion assays, Nap1 purification, and microscope surface cleaning/prep in the Methods section.

Reviewer #3 (Remarks to the Author)

The manuscript by Gao et al characterizes the role of the histone chaperone Nap1 as well as histone H3 and H4 acetylations in regulation of liquid liquid phase separation (LLPS) of chromatin employing a 12-mer 601 Widom nucleosome array. They identify the histone H4 tail as a primary driver of chromatin liquid-liquid phase separation, with a less significant but a contributing role for histone H3 tails. They present and discuss the contribution of H3 and H4 tail acetylation in mediating the chromatin phase separation. They further show that the histone chaperone Nap1 plays a role in regulating the condensate in resolving aggregates and enhancing fluidity of the condensates. They also demonstrate that there is a scaffold of relatively static chromatin elements within the chromatin droplets. Finally, they show that

chromatin droplets with acetylation mimic and Nap1 has a lower viscosity. The paper adds important additional details to the understanding of the mechanisms of chromatin LLPS and should be published after major revision.

We appreciate the constructive comments from the reviewer. We addressed all the comments as detailed below.

Comments:

- Generally linguistic review would help to improve the text. See some examples below
- The title should use the more well-defined term “nucleosome array” not “chromatin array”. Same in the abstract where this is used once. In the rest of the text they correctly use “nucleosome array”.

The term “chromatin array” has been changed to “nucleosome array”.

- “Chromatin condensation is dynamically regulated throughout the cell cycle and plays key roles in modulating gene accessibility at the highest physical level in a cell.”: Suggests rephrasing “highest physical level”, it is unclear what this means.

This phrase has been removed in this revision.

- The fundamental “gene packing” unit in chromatin is the nucleosome, which is ~150 bp DNA wrapped around an octameric histone core containing two H2A-H2B dimers and one (H3-H4)₂ tetramer. “DNA packaging” is more accurate.

The suggested change has been made in this revision.

- “...undergo liquid-liquid phase separation (LLPS) to form liquid-like droplet condensates 27,28.” Relevant missing reference: <https://doi.org/10.3390/cells11193145>

The suggested reference has been added to this revision.

- “spherical, suggesting that the droplets begin in a liquid-like state and age into a gel-like state 29.” The findings of this paper are largely disproved by reference 27 that proposed that the observations may be due to sample preparation issues. Hence, conclusions should be taken with a grain of salt.

The suggested information has been added to this revision as follows in page 2:

“Moreover, the gel-like nature of the droplets has been largely disproved by a recent report²⁷.”

- “acetylations on H3 or H4 have been implicated in weakened intra- and inter-nucleosomal DNA-histone interactions 30–35” A key references missing: <https://doi.org/10.1093/nar/gkq900> Another key reference missing here or when tailless construct results are discussed is: doi: 10.1016/s0022-2836(03)00025-1

The suggested references have been added to this revision.

- Methods: There are no details on size exclusion chromatography of 12-mer DNA and assembled arrays and analytical ultra centrifugation. It is strongly suggested adding these details.

Details of size exclusion chromatography of DNA and arrays, and analytical ultracentrifugation have been added to this revision in the Methods section and in the Supplementary Information (Figs. S3 and S4).

- Figure S1. An agarose gel image of fractions showing the separation of the DNA would be a good addition.

An agarose gel image showing the plasmid digestion product before and after FPLC purification has been added to Fig. S1 in this revision (Fig. S1).

- Figure S2. It would be ideal to indicate the ratio for each band in the gel image and also the determined optimal saturation ratio in the gel. Is the loading of chromatin the same for all gel panels? It appears that the amount of competitor DNA used is different for the different ratios unless this is an artifact due to different gel run times.

The suggested information on the histone:DNA ratios for the 5 types of the array samples have been added to the revision (Methods and Fig. S2). The amounts of arrays and competitor DNA are constant across all lanes in each sample. The reason why the competitor DNA band becomes dimmer at a higher histone:DNA ratio is because the competitor DNA forms aggregates of various sizes with randomly loaded excess histone after array DNA is saturated. This information has been added to Fig. S2 caption.

- Comment on size exclusion chromatography purification of assembled arrays: This is not a widely employed protocol though preparative methods are used to separate free DNA and mononucleosome. An agarose gel for the SEC purified chromatin sample and SDS-gel verification of the presence of 4 histones is recommended. In S3, there are significant differences in the sedimentation rate of different arrays. Based on the findings of the manuscript one would expect H4KQ and gH4 to have the most open chromatin however this is not reflected in the AUC results. SDS analysis is suggested of the chromatin to ensure there was no dimer loss during the size exclusion chromatography of the chromatin, especially for the gH3 sample owing to its lower stability. As the authors have established a protocol for TEM imaging, alternative to SDS/gel an analysis of the mutant and globular arrays that have gone through SEC would also suffice.

We agree that the AUC results may not show the actual size and composition of the nucleosome arrays because the chain conformation and the charge distribution surrounding a nucleosome array depend on many parameters that cannot be precisely analyzed solely with an AUC curve. Instead, as the reviewer suggested, we added an agarose gel for the SEC purified samples and an SDS PAGE gel to show the presence of all 4 histones in those samples in the revision (Fig. S4c:WT array before digestion, and Fig. S4d). We went one step further for the agarose gel analysis. We used a restriction enzyme to show that the nucleosomes in the arrays are properly wrapped (Fig. S4c). These analyses clearly show proper saturation of the nucleosome arrays with histone. The results and the explanations have been added to the Methods section and Fig. S4.

- “Among these four samples, only the H3KQ arrays result in chromatin droplets (Fig. 2a).”: A comment on comparison with wild type would be informative to the reader.

In this revision, we tested the extent of phase separation and droplet formation at varying concentrations of the arrays (25, 50, 100, and 200 nM) and NaCl (5, 50, 100, and 150 mM) utilizing UV-Vis absorbance measurements, and constructed phase diagrams of all cases shown in Figs. 1 and 2. Droplets scatter light at wavelengths according to their sizes. The scattering extent in a range of wavelengths (350 – 840 nm) was measured as absorbances in a UV-Vis spectrometer. Non-zero detectable absorbance in this range confirms the formation of droplets or aggregates down to a few hundred nanometers in size. The abundances of the droplets and aggregates of various sizes measured as the integrated absorbance (340 – 840 nm) are presented as phase diagrams (Fig. 1b and Fig. 2bcd). The phase diagrams clearly show less efficient droplet formation with H3KQ arrays than WT arrays at given array and NaCl concentrations. The H3KQ arrays require a higher array concentration and/or a higher NaCl concentration to form droplets than WT arrays. The new data and explanations have been added to Figs. 1 and 2 and the main text (pages 11-12):

“To survey the extents of droplet formation under a range of conditions, the UV-Vis absorbances of the arrays were measured at varying array concentrations (25, 50, 100 and 200 nM) and NaCl concentrations (5, 50, 100 and 150 mM). Droplets scatter light at wavelengths according to their sizes. The scattering extent was measured as a UV-Vis absorbance spectrum within the range of 350 – 840 nm. Non-zero detectable absorbances in this range confirm the formation of droplets down to a few hundred nanometers in size. A time series of changes in the UV-Vis spectrum during the first 30 minutes of droplet formation is shown in Fig. S5. As the changes are not monotonic increase or decrease at a specific wavelength in this range, we used the integrated UV-Vis absorbance (350 – 840 nm) after 30 minutes of droplet formation as a measure of the abundance of the droplets of various sizes. The measured droplet abundances are presented as a phase diagram (Fig. 1b).”

“The phase diagrams (Figs. 1b and 2b) clearly show less efficient droplet formation with H3KQ arrays than WT arrays at given array and NaCl concentrations, and no droplet formation with H4KQ arrays.”

- Page 11: “Upon a closer look at the droplets, we observed crumbling from the inside, loss of the round outer fringe, and shrinkage of the droplet size within 10 – 15 minutes”: An image would be ideal, please add in supplementary.

We added a time series of a droplet showing crumbling, losing its round shape, and shrinking in Fig. S8.

- Page 11: “We hypothesize that the aggregation is caused by nucleosome disassembly upon nucleosome destabilization due to the lack of the H3 tails.” A TEM imaging of the arrays might show evidence for this, in form of aggregates or disassembled nucleosomes.

- “These results validate our hypothesis of disassembled nucleosomes and random histone-DNA mixture in condensed chromatin containing tailless H3.”: The disassembly of nucleosomes was not confirmed with direct/indirect observation; hence the above statement is not accurate.

Comment: Nucleosomes are sensitive to cations, and this was exploited in obtaining nucleosome crystals. The crystallization experiments often resulted in precipitation/aggregates that probably might not have been due to disassembly. The aggregates observed might possibly be trapped nucleosome-nucleosome interaction that might not be the most energetically favored interaction. The Nap1 probably might just be ‘releasing the trapped nucleosomes’ and facilitating the most energetically favorable nucleosome-nucleosome interactions. However, if the authors can validate the disassembly, it would be very nice.

It is not possible to take clear images of disassembled and entangled arrays with TEM. Instead, we show the extent of partial disassembly of nucleosomes in gH3 array aggregates with two restriction enzymes (Fig. S10). The 601 nucleosome sequence contains a HinfI restriction site at the 5th – 10th nt from the entry and a BsrBI restriction site at the 23rd to 28th nt from the entry. We digested the aggregated gH3 arrays and analyzed them on an agarose gel. The gel shows clear digestion by HinfI, but not by BsrBI, showing partial opening of the nucleosomes near the entry/exit region. Treatment of the aggregates with Nap1 makes the arrays digested much less by HinfI, confirming that Nap1 catalyzed re-assembly of the nucleosomes. We added Fig. S10 and the following explanations in the main text (page 13):

“To examine the extent of nucleosome disassembly in gH3 arrays, we employed restriction digestion assays with two restriction enzymes HinfI and BsrBI (Fig. S10). The 601 nucleosome sequence contains a HinfI restriction site at the 5th – 10th nt from the entry and a BsrBI restriction site at the 23rd to 28th nt from the entry. Partially unwrapped nucleosomes at the nucleosome termini will make the HinfI site accessible and digested by the enzyme while more completely disassembled nucleosomes will make the BsrBI site accessible and digested. Our digestion assay shows that the gH3 arrays are digested by HinfI while they are not digested by BsrBI, supporting potential partial disassembly of the nucleosomes near the entry/exit region in the aggregates (Fig. S10). Upon Nap1 treatment to dissolve most of the aggregates, we see a much lower level of digestion by HinfI, confirming that some of the partially unwrapped nucleosomes are re-wrapped.”

- The results in the later part though reasonable and logical, would be convincing if backed with a thorough analysis of the chromatin sample preparation.

We added an agarose gel with restriction enzyme treated arrays and an SDS-PAGE gel to show proper histone saturated array samples (Fig. S4). Please see the above for more information about the sample preparation and analyses.

- The preparation of droplets by addition of phase separation buffer:

The details of the procedure are not provided; hence the query is: is the concentration of salt in the buffer very high in order to induce dissociation in the gH3 and gH4 samples? Have the gH3 arrays been confirmed not to have dimer loops during SEC? 601 nucleosomes have relatively high stability, apart from reported breathing and dynamics in literature, the nucleosome requires high molarity salt to induce dissociation.

The stock buffer concentration that is added to the samples to induce phase separation is 200 – 300 mM NaCl, not high enough to disassemble nucleosomes instantaneously. We show with SDS PAGE that gH3 arrays have all 4 histones (Fig. S4).

Reviewer #2 (Remarks to the Author):

The authors have addressed most of my concerns. There are a few additional points to be considered before recommending the manuscript for publication in Nature Communications:

1. The extent of acetylation and probably also the precise acetylated sites of H3 catalyzed by Ada2/Ada3/Gcn5, and H4 by Piccolo NuA4 should be accessed with proper mass spectrometry techniques. This information is essential to well support the conclusions about the roles of different acetylation in the phase behaviors.

Quantifying acetylation extents at multiple sites within a single protein embedded in a large complex is challenging due to difficulties in reliable peak normalization and other analytical limitations. Therefore, we revised the manuscript to note that we cannot attribute the observed effect of H3 acetylation to specific lysine residues or to the extent of their modification. Nonetheless, our results clearly demonstrate that H3 acetylation enhances DNA-histone dynamics in condensates. The text (page 8) reads:

“We then repeated the FRAP assay with the droplets containing arrays that are H3-acetylated (H3ac) by this HAT complex (Fig. 4a). The droplets display considerably faster FRAP recovery than those containing WT arrays, while the recovery fraction remains unchanged. The HAT complex has a similar acetylation activity to the entire SAGA complex, which acetylates mainly H3 tail lysine residues, including K9, K14, K18, and K23^{57,58,59}. However, the exact locations and extents of acetylation by Ada2/Ada3/Gcn5 are yet to be clearly defined. Consequently, we cannot attribute the observed effect of H3 acetylation to specific lysine residues or to the degree of their modification. Nonetheless, these results confirm that H3 acetylation considerably enhances DNA-histone dynamics in condensates.”

2. Typo in Fig 2 caption, “clearing showing a similar.....”

The typo is corrected in Fig. 2 caption.

Reviewer #3 (Remarks to the Author):

The manuscript has significantly improved with the additions and amendments. The major conclusions of H4 being primary driver, role of Nap1, mobile and structural scaffold are a welcome addition to our understanding of chromatin phase separation.

Some minor comments:

Some review of certain phrase is still strongly recommended to improve readability and clarity of the ideas being presented. Few examples are listed below:

Abstract:

Suggest reviewing the following as it is difficult to comprehend:

“We also show that the condensed liquid-like droplets contain both a mobile fraction and a relatively immobile structural scaffold and that histone chaperone Nap1 and histone H3 tail acetylation facilitate DNA-histone dynamics within the structural scaffold to lower the overall viscosity of the droplets.”

This sentence is split into two for clarity. The text reads (Abstract):

“We also show that the condensed liquid-like droplets comprise a mobile fraction and a relatively immobile structural scaffold. Histone chaperone Nap1 and histone H3 tail acetylation enhance DNA-histone dynamics within this scaffold, thereby lowering the overall viscosity of the droplets.”

Introduction “The structure of chromatin is formed largely by arrays of genomic DNA, histone proteins, and various other chromatin-associated proteins.”

The structure of chromatin is still debated; “chromatin is composed of” is more appropriate.

The suggested change has been made (page 2).

74:Suggest to amend or refrain from claim of primacy. Ref 37 reports effects of H4 tail acetylation on chromatin folding and self-association, which is analogous to condensation behavior of chromatin arrays.

We revised the text to avoid claim of primacy. The text (page 3) reads:

“However, the effects of specific histone tail acetylations on the condensation behavior of nucleosome arrays remain poorly understood.”

367 ‘at physiological nucleosome concentration’ might not be accurate. The nucleosome concentration in the experiments is 200 nM, though it reaches much higher concentration approaching physiological concentration within the droplets.

This phrase has been removed. The text (page 5) reads:

“To further confirm the effect of H4KQ, we tested if H4 tail acetylation with histone acetyltransferase (HAT) also inhibits LLPS. We employed the Piccolo NuA4 complex...”

509-520 Some of the details are a better fit within methods

We moved the details to the Methods section. The remaining text (page 10) reads:

“We took fluorescence images of chromatin droplets containing Tet-labeled nucleosome arrays and constructed their STORM images (see Methods). The 10 s Total STORM images in Fig. 5 show all fluorescent nucleosome arrays detected during a 10 s imaging period. Within each STORM image, we identified Clusters of nucleosome arrays that remained immobile during this period within a 110 nm radius, approximately the contour length of a 12-mer nucleosome array. The 10 s Clusters STORM images in Fig. 5 display these immobile arrays.”

Nap1 increasing both nucleosome concentration and fluidity is fascinating. A comment on its physiological relevance would be good.

We added a brief point of discussion on page 16 about the potential physiological relevance of the Nap1’s role. The text (page 16) reads:

“Therefore, Nap1 likely promotes a dynamic yet condensed chromatin environment that facilitates nucleosome turnover and regulatory protein access. This property may underlie Nap1’s physiological role in maintaining chromatin plasticity during processes such as transcription, DNA replication, and chromatin remodeling.”